# Genomic dissection of an extended phenotype: Oak galling by a cynipid gall wasp

Jack Hearn[1,2]*, Mark Blaxter[2], Karsten Schönrogge[3], José-Luis Nieves-Aldrey[4], Juli Pujade-Villar[5], Elisabeth Huguet[6], Jean-Michel Drezen[6], Joseph D. Shorthouse[7], Graham N. Stone[2]*

**1** Vector Biology Department, Liverpool School of Tropical Medicine, Liverpool, United Kingdom, **2** Institute of Evolutionary Biology, University of Edinburgh, King's Buildings, Edinburgh, United Kingdom, **3** Centre for Ecology and Hydrology, Wallingford, United Kingdom, **4** Departamento de Biodiversidad y Biología Evolutiva, Museo Nacional de Ciencias Naturales (CSIC), Madrid, Spain, **5** Departament de Biologia Animal, Universitat de Barcelona, Spain, **6** UMR 7261 CNRS, Institut de Recherche sur la Biologie de l'Insecte, Faculté des Sciences et Techniques, Université de Tours, France, **7** Department of Biology, Laurentian University, Sudbury, Ontario, Canada

* Jack.Hearn@lstmed.ac.uk (JH); Graham.Stone@ed.ac.uk (GNS)

## Abstract

Galls are plant tissues whose development is induced by another organism for the inducer's benefit. 30,000 arthropod species induce galls, and in most cases the inducing effectors and target plant systems are unknown. Cynipid gall wasps are a speciose monophyletic radiation that induce structurally complex galls on oaks and other plants. We used a model system comprising the gall wasp *Biorhiza pallida* and the oak *Quercus robur* to characterise inducer and host plant gene expression at defined stages through the development of galled and ungalled plant tissues, and tested alternative hypotheses for the origin and type of galling effectors and plant metabolic pathways involved. Oak gene expression patterns diverged markedly during development of galled and normal buds. Young galls showed elevated expression of oak genes similar to legume root nodule Nod factor-induced early nodulin (*ENOD*) genes and developmental parallels with oak buds. In contrast, mature galls showed substantially different patterns of gene expression to mature leaves. While most oak transcripts could be functionally annotated, many gall wasp transcripts of interest were novel. We found no evidence in the gall wasp for involvement of third-party symbionts in gall induction, for effector delivery using virus-like-particles, or for gallwasp expression of genes coding for plant hormones. Many differentially and highly expressed genes in young larvae encoded secretory peptides, which we hypothesise are effector proteins exported to plant tissues. Specifically, we propose that host arabinogalactan proteins and gall wasp chitinases interact in young galls to generate a somatic embryogenesis-like process in oak tissues surrounding the gall wasp larvae. Gall wasp larvae also expressed genes encoding multiple plant cell wall degrading enzymes (PCWDEs). These have functional orthologues in other gall inducing cynipids but not in figitid parasitoid sister groups, suggesting that they may be evolutionary innovations associated with cynipid gall induction.

**Data Availability Statement:** Raw Illumina Hi-Seq data have been deposited in the European Nucleotide Archive under BioProjects PRJEB13357, PRJEB13424, and PRJEB32849.

**Funding:** This study was supported by a NERC studentship awarded to Jack Hearn, a NERC NBAF grant (NBAF375 2009-2010) awarded to Graham Stone, and NERC grant NE/J010499. Jose-Luis Nieves Aldrey was supported by project CGL2015-66571-P (MINECO, Spain).The funders had no role in study design, data collection and analysis, decision to publish, or preparation of the manuscript.

**Competing interests:** The authors have declared that no competing interests exist.

## Author summary

Plant galls are induced by organisms that manipulate host plant development to produce novel structures. The organisms involved range from mutualistic (such as nitrogen fixing bacteria) to parasitic. In the case of parasites, the gall benefits only the gall-inducing partner. A wide range of organisms can induce galls, but the processes involved are understood only for some bacterial and fungal galls. Cynipid gall wasps induce diverse and structurally complex galls, particularly on oaks (*Quercus*). We used transcriptome and genome sequencing for one gall wasp and its host oak to identify genes active in gall development. On the plant side, when compared to normally developing bud tissues, young gall tissues showed elevated expression of loci similar to those found in nitrogen-fixing root nodules of leguminous plants. On the wasp side, we found no evidence for involvement of viruses or microorganisms carried by the insects in gall induction or delivery of inducing stimuli. We found that gall wasps express many genes whose products may be secreted to the host, including enzymes that degrade plant cell walls. Genome comparisons between galling and non-galling relatives showed cell wall-degrading enzymes are restricted to gall inducers, and hence potentially key components of a gall inducing lifestyle.

## Introduction

Galls are plant tissues whose development is induced by other organisms, including viruses, bacteria, protozoa, fungi, nematodes, mites and insects [1,2]. Galls are not simply wound responses, but are specific plant growth responses to infection that provide the inducing organism with nutrition and protection from external abiotic and biotic challenges [1,2]. Galls represent sinks for plant nutrients and metabolites, diverting resources towards gall inducer development that could otherwise be invested in plant maintenance, growth and reproduction [3–6]. While some galling interactions are clearly mutualistic (e.g. legume root nodules induced by nitrogen-fixing *Rhizobium* bacteria [7] and the galls induced in figs by their pollinating wasps [8–10]), most have no apparent benefit for the host plant. That the impact of galling can be strongly negative for the host plant is shown by the fact that gall inducers include many serious pests of plants in agriculture and forestry [11–13], and important biocontrol agents of invasive plants [14,15].

The molecular mechanisms underpinning gall induction are only well understood in a small number of microbial systems, including *Rhizobium* induction of root nodules, crown gall induction by *Agrobacterium tumefaciens*, and the parasitic fungus *Ustilago maydis* of maize [7,16,17]. These systems are characterised by (i) release of host plant molecules that are detected by the gall inducing organism and (ii) release by the gall inducer of molecules, termed effectors, that (iii) interact with specific plant receptors. These interactions (iv) result in characteristic plant growth responses to produce gall tissues.

In contrast to microbial systems, gall induction by animals is poorly understood for all but a tiny minority of economically important species, including root knot and cyst nematodes (*Heterodera*, *Globodera* and *Meloidogyne* species) [18–20], Hessian fly (*Mayetiola destructor*) [21–24] and *Phylloxera* [25,26]. Nevertheless, with an estimated 30,000 species [27], gall inducing arthropods (and particularly insects) are abundant and ecologically important components of many biological communities [27]. A key feature of insect-induced galls is that, relative to microbial galls, most are structurally both more defined and more complex, with clear differentiation between distinct tissues associated with inducer nutrition and protection [1,28,29].

Because gall traits are characteristic of the inducing species rather than the plant host, Dawkins [30] and many others [26,31–34] have viewed plant galls as the extended phenotypes of gall inducer genes, stimulating research into the adaptive significance of insect gall morphologies. Similarity in the gall morphologies induced by phylogenetically divergent insect lineages also raises the question of whether different inducers use similar effectors to target similar plant systems to reset or divert normal plant development [6,26,35,36].

The oak gall wasps (Hymenoptera: Cynipidae: Cynipini) are one of the most species-rich lineages of galling insects, with *ca.*1500 species that induce diverse and structurally complex galls on oaks (*Quercus*) and related Fagaceae [37–39]. The lifecycles of most oak gall wasps involve obligate alternation between a sexual and an asexual generation, each of which induces a different gall phenotype on a specific organ (leaf, bud, flower, fruit, shoot or root) on specific host Fagaceae [40,41]. Evolutionary shifts between host plant lineages have been extremely rare in gall wasps [39,42]. These patterns suggest that the association between gall wasps and their hosts is physiologically intimate, involving gall wasp manipulation of specific plant developmental pathways. Several properties of oak-gall wasp interactions make this system an interesting one for exploration of processes associated with gall induction and development: (i) the developmental anatomy of some oak galls has been studied in considerable detail, allowing division of gall development into recognisable stages [1,28,41,43–46] between which patterns of gene expression can be compared; (ii) genomic resources are increasingly available for some oaks [47,48] and oak gall wasps [49–51]; (iii) oak gall diversity makes this an excellent system in which to study the adaptive significance of gall phenotypes [29,38,52,53]; and (iv) the ancestral life history strategy of cynipids prior to evolution of gall induction is known. Gall wasps evolved from parasitoids similar to extant figitid cynipoids, which develop within the bodies of herbivorous insects living within plant tissues [39,54,55]. Genomic comparisons between cynipid gall inducers, and between cynipids and their figitid sister group, thus provide a means to identify innovations associated with diversification of gall phenotypes and the origins of gall induction.

Initiation of cynipid gall induction begins when a female oviposits into a specific location in meristematic tissues on a specific host plant [40,41,53]. When the first instar gall wasp larva (*ca*. 0.3mm long) hatches from the egg, the surrounding plant cells disintegrate and/or de-differentiate to form a chamber lined with callus-like tissue into which the larva moves [1,28]. While initiation of cynipid gall development may involve maternal secretions in the egg [1,49], it has long been known that if the gall wasp larva is killed, gall development ceases, implying that larval production of inducing effectors is required for continued gall growth [41,56]. Cynipid gall development following initiation can be divided into three stages—Early, Growth, and Mature [28,57]—that represent natural sampling points for associated processes. These stages are illustrated for the sexual generation gall induced by our study species, *Biorhiza pallida*, on leaf buds of pedunculate oak, *Quercus robur*, in Fig 1. This gall type was selected because its developmental anatomy [1,44,45,58] and physiology [59,60] have been relatively well studied, and also because a single founding female lays 50–400 asexually-produced eggs into a single bud [61,62]. The resulting larvae develop in synchrony, facilitating the detection of transcripts from multiple larvae of the same age from whole gall RNA extractions. In Early stage galls (Fig 1A, 1B and 1C), larvae are < 1 mm long. Host tissues lining the larval chamber divide rapidly and differentiate into specialised nutritive cells that provide all larval food [63]. A thin wall of sclerenchyma separates each inner gall (larval chamber) from non-nutritive outer gall tissues [37,43,63,64]. In Growth stage galls (Fig 1A, 1D and 1E) the larvae remain very small, but development and differentiation of outer gall tissues is extensive [1,65]. The nutritive cells multiply (hyperplasy) and enlarge (hypertrophy), become multinucleate, undergo endoreduplication of their chromosomes, and accumulate high concentrations of lipids, carbohydrates

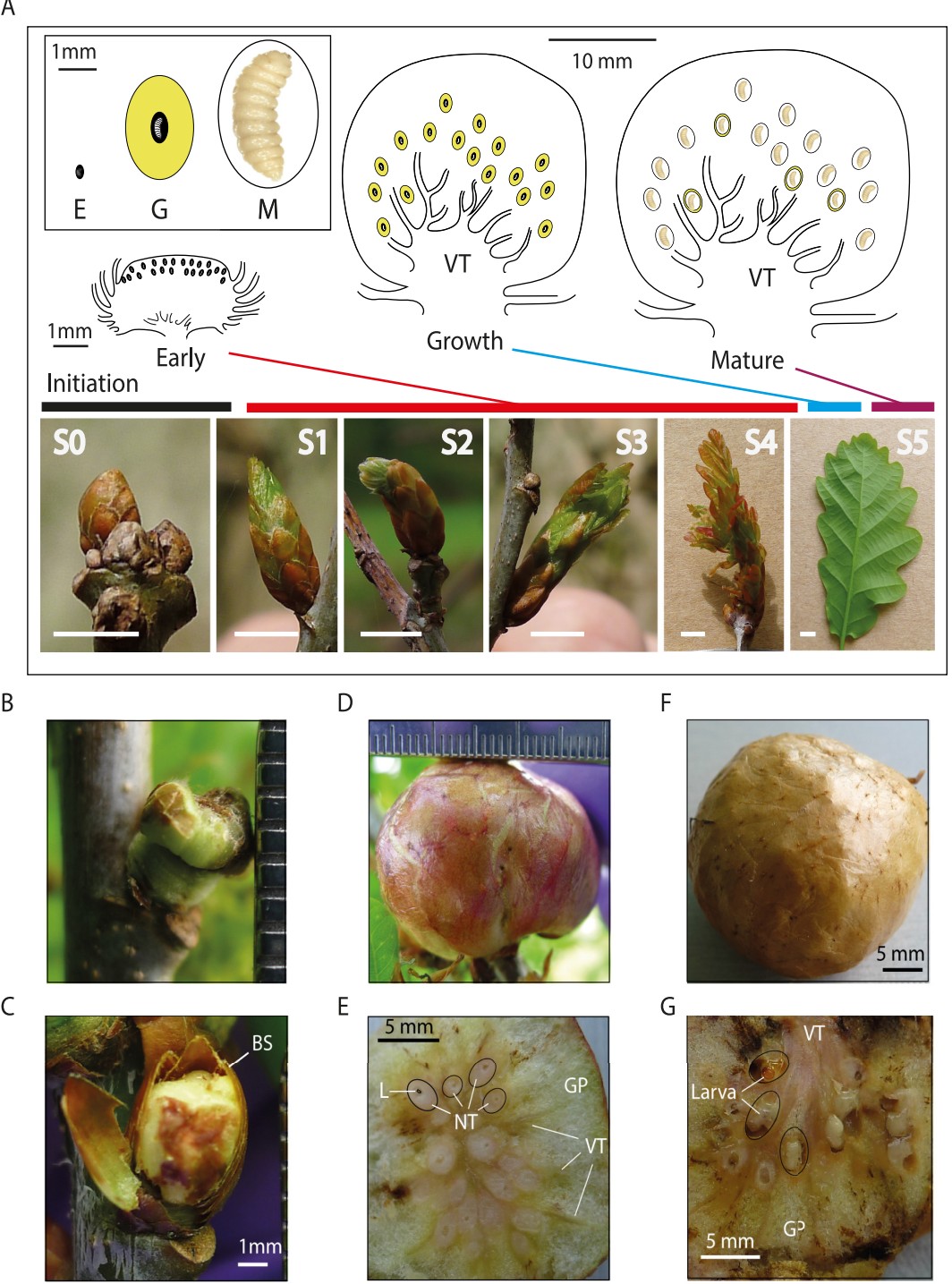

**Fig 1. Stages in the development of the sexual generation galls of *Biorhiza pallida* and of ungalled buds of *Quercus robur*.** A: Diagrammatic sections of Early, Growth and Mature stage galls, and their developmental phenology relative to recognised stages in the development of normal buds. Full gall development takes between 3 and 6 weeks. The inset shows the relative sizes of the larval chambers, larvae and nutritive tissues in each gall stage, with nutritive tissues shaded in yellow, fluid filled space in black, and an air space in white. Normal bud development in *Quercus robur* is shown for sampled examples of six widely recognised developmental stages: S0—Buds appear dormant with no bud-burst activity; S1—buds have the first visible swelling of bud-burst; S2—buds have started to grow as indicated by increased bud length and diameter; S3—distinct leaves are now visible; S4—leaf and stem development has begun; S5 leaves are now mature, and the associated stem has begun to elongate. Scale bars in panels S0-S5 are

5mm. B-G: Whole galls (B,C,D,F) and sections (E,G) of galls of each stage sampled in this study. B, C: Early stage galls are less than 5 mm in diameter, and are commonly partially concealed by bud scales (BS; removed in B, partially removed in C). The gall contains multiple larval chambers, each containing a single gall wasp larva. At this stage, the gall wasp larvae are *ca*. 0.25 mm long. D, E: Growth stage galls are 20–40 mm in diameter, with clearly differentiated tissues in section. The red epidermis is characteristic of these galls on exposure to sunlight. The sectioned gall (E) shows the spongy gall parenchyma (GP) developing around the larval chambers, four of which have been outlined in black. Each larval chamber at this stage contains nutritive tissues (NT) surrounding a larva 0.25–1.0 mm long (an example is outlined in black, and labelled L). Vascular tissues (VT) can be seen radiating through the gall from the central point of connection with the oak shoot. F, G: Mature stage galls have a brown and papery epidermis. The sectioned gall (G) shows large feeding larvae within their fully developed chambers, with pronounced lignification of surrounding tissues. In the upper outlined chamber all of the nutritive tissues have been consumed, and the head capsule of the mature larva is visible.

and nitrogenous compounds [44,45,59,60,62,63,66]. Outside the nutritive core, gall growth proceeds rapidly by hyperplasy and hypertrophy of layers of parenchyma [37,46]. Extensive development of vascular tissues facilitates import of nutrients and water from the host (Fig 1A, 1E and 1G). In the transition to Mature stage galls (Fig 1A, 1F and 1G), the larvae grow rapidly until they exhaust the nutritive tissues and reach the surrounding sclerenchyma layer. Tissues outside the larval chamber become lignified and desiccated. The larvae then pupate and adult *B. pallida* emerge two to three weeks later.

Normal (ungalled) buds develop very differently (Fig 1A), through six developmental stages recognised in previous work on *Quercus robur* and other trees [67] that span the transition from dormant buds (S0) to fully open leaves (S5). The sexual generation galls of *Biorhiza pallida* develop [1,28] slowly in comparison to ungalled buds, such that fully flushed leaves develop from ungalled buds in the same time taken for galled buds to reach the Early and Growth stages (Fig 1A).

Very little is known about the plant metabolic processes underlying cynipid gall development, the nature of the effectors involved, or their source(s) [68–70]. The aim of this study was to provide the first characterisation of oak and gall wasp gene expression in gall development immediately after initiation, and to discriminate between alternative hypotheses for the source(s) of gall wasp-associated effectors. To identify plant processes involved through gall development, we identified oak genes with strong stage-specific patterns of differential expression in Early, Growth and Mature stage galls. To identify differences between galled and normally developing tissues, we identified oak genes with contrasting expression trajectories in development of galled and ungalled (control) buds. We identified gall wasp genes with strong stage-specific patterns of differential expression in the same gall stages, and used functional annotation to infer their potential roles in gall wasp development and interaction with their oak host. Finally, we use *de novo* genomic data for a panel of gall inducing cynipids and parasitoid figitids to identify novel gall wasp genes associated with a gall inducing life history. We use these approaches to address four general questions in cynipid gall development, outlined in detail below.

## What, in plant terms, is a cynipid gall?

Cynipid galls include multiple, organized tissue types, and can be considered novel plant organs [59,60]. We tested two existing and mutually compatible hypotheses based on previous work (see Discussion): 'galls as ectopic food storage organs', and 'galls as modified somatic embryos' (Fig 2A) [59,60]. The first hypothesis stems from the observation that cynipid gall nutritive tissues express biotin carboxylase carrier protein (*BCCP*) [59,69], a component of the triacylglycerol lipid synthesis pathway associated with lipid food storage in seeds and tubers [59,60,70]. Cynipids could manipulate lipid metabolism to enhance the development of tissues on which the larvae feed, and galls could thus be modelled as ectopic plant food storage organs.

The second hypothesis stems from the observation that Early stage cynipid galls display cellular dedifferentiation and rapid cell proliferation [45,59,60], phenomena observed during normal somatic embryogenesis [71]. Cynipid galls may thus also be modelled as developing from ectopic somatic embryos (Fig 2A). We explored the nature of plant-derived gall tissue by transcriptomic analysis, in particular seeking to identify genes and systems typical of storage organs and somatic embryos that were highly (or only) expressed in Early *versus* later gall tissues, and genes whose trajectory of expression through gall development contrasted with patterns observed in normally developing oak buds.

## What organism induces the *B. pallida* oak gall?

The observation that cynipid gall development requires a living larva [43,56] could indicate endogenous gall wasp production of effectors required for gall development. However, the same observation could indicate a key role for one or more endosymbionts (viruses, bacteria or other microorganisms [72]) in effector production and/or delivery (Fig 2B). Host plant

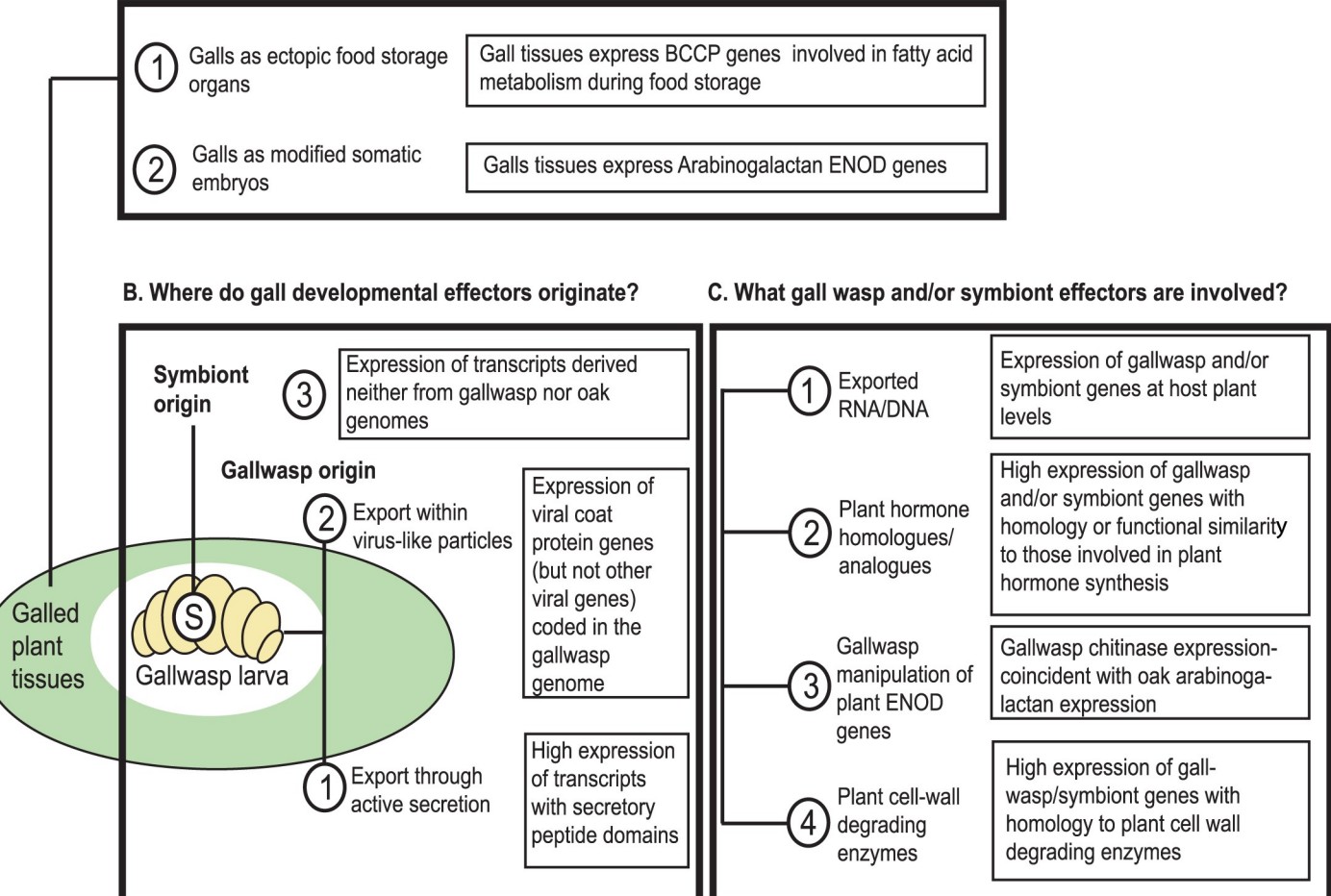

**Fig 2. Patterns of oak, gall wasp and symbiont gene expression expected under alternative hypotheses for cynipid gall development.** A. Oak gene expression patterns predicted under the ectopic food storage organ and somatic embryo hypotheses. B. Alternative hypotheses for the origin(s) and delivery of gall development effectors. C. Alternative possible effectors. In each of A-C, boxed texts summarise expectations for patterns in transcriptome data if the hypotheses are supported (see main text Introduction and Discussion for explanation). The circled '**S**' within the gall wasp larva represents a possible internal symbiont, such as a virus or bacterium.

manipulation by a *Wolbachia* symbiont has been shown for a leaf mining moth [73], and bacterial symbionts are required for successful gall development in several non-cynipid systems [74–76]. Symbionts can fulfil other crucial roles in gall tissues—for example, the larvae of ambrosia gall midges (Diptera; Cecidomyiidae) feed on fungal symbionts whose spores are inoculated into host plant tissues by the ovipositing female [77,78]. Microbial symbionts could thus be involved in cynipid gall induction, development or physiology. We hypothesised that involvement of coinfecting and gall wasp-transmitted symbionts should be associated with repeatable detection of (i) symbiont genomes in assemblies generated for female gall wasps, and (ii) symbiont transcripts in transcriptomes for relevant gall developmental stages.

We also assessed the potential for gall wasps to use genes derived from viral symbionts to deliver endogenous or symbiont effectors (Fig 2B). Parasitoid wasps export DNA or proteins that suppress insect host immune responses [79,80] by packaging them within virus-like-particles (VLPs) [79,81], the components of which are coded for by viral genes that have been incorporated into the parasitoid genome [80,81]. *Leptopilina* species in the Figitidae sister group of cynipids use VLPs to deliver host immune-suppressing factors [82], and gall wasps could in principle use a similar delivery system. Involvement of VLPs in signal delivery should be associated with detection of virally-derived genes in the gall wasp genome (as indicated by flanking of putative VLP sequences by unambiguously insect genes and/or presence of introns) and detection of VLP transcripts during gall development.

## What processes are involved in gall wasp manipulation of plant development?

Gall wasp larvae remain very small during much of gall growth, and the assumption for over a century [83] has been that they excrete effectors into the liquid medium that surrounds them early in gall development (Fig 1A). Gall wasp larvae have a sealed anus and enlarged salivary glands, making salivary (and potentially also Malpighian tubule) secretions plausible candidates for the source of induction effectors [83,84]. Gall development could be induced through gall wasp (or associated symbiont) manipulation of plant hormone cascades (by secretion of hormone homologues or analogues, or enzymes that modulate host small molecule metabolism), secretion of direct protein regulators of plant developmental processes, and/or secretion of enzymes that change the physiology of plant cells (Fig 2C, and see Discussion). We used gall wasp genomic and larval transcriptome data to identify gall wasp genes with potential to interact directly with plant metabolism, and that were differentially expressed between Early and later gall stages. We also sought to identify parallel expression patterns in oak genes involved in hormone transport, receptors and target systems, and to identify gall-specific processes by comparison of expression trajectories in gall and normal (non-galled) oak tissues.

## Which gall wasp genes involved in gall development are novel compared to non-galling sister groups?

We hypothesise that, since diverging from a parasitoid common ancestor shared with their Figitidae sister group approximately 80 million years ago [55], gall wasps will have evolved shared novel traits associated with herbivory and host plant manipulation. We therefore looked for homologues of *Biorhiza pallida* genes that were highly expressed during gall development in the genomes of exemplar species in six additional gall-inducing cynipid tribes and two species of non-galling parasitoid Figitidae. Our rationale is that genes that are highly expressed during gall development and shared by gall wasps but absent from their parasitoid sister groups represent candidates for evolutionary innovations associated with a gall-inducing lifestyle.

We show that while gall tissues of the sexual generation of *Biorhiza pallida* share some similarities with normally developing buds, patterns of oak gene expression show consistent differences between these two structures. We find no evidence for involvement of known symbionts in cynipid gall induction, or for export of gall wasp gene products in virus-like particles. We find the *Biorhiza pallida* genome to encode a suite a plant cell wall degrading enzymes, including cellulases, that are expressed at various points through gall development. These genes are shared with a phylogenetically diverse panel of gall wasps but are absent from non-galling sister groups, suggesting that they are shared evolutionary novelties associated with a gall inducing life history. Finally, we develop a hypothesis for chitinase-mediated gall wasp modification of plant arabinogalactan molecules during gall development.

## Results

### Generation of RNA-seq datasets for *B. pallida* galls and ungalled oak buds

We performed dual-RNA-seq [85] on four biologically independent replicates for each of Early, Growth and Mature stage galls (Fig 1; S1 Fig) and also for each of five of the six developmental stages of normally-developing ungalled pedunculate oak (*Q. robur*) buds from dormant buds (S0) through bud expansion (S1, S3, S4) to fully-opened leaves (S5) [47,67] (Fig 1G). Raw Illumina Hi-Seq data have been deposited in ENA under BioProjects PRJEB13357, PRJEB13424, and PRJEB32849. Detailed descriptions of sequence data processing are provided in Materials and Methods.

We generated a co-assembly of all the gall and non-gall RNA-seq data and then separated and filtered genes (component groups in TRINITY terminology) by taxonomic origin to identify transcripts deriving from oak, gall wasp, parasitoid, viral and fungal origins. We used a genomic approach to seek potential bacterial symbionts as the poly-A selection protocol we used is unlikely to detect bacterial transcripts (see below). The combined assembly spanned 592 megabases (Mb) in 773,991 transcripts with an N50 of 1,145 bases. Further filtering to remove ribosomal RNA sequences and retain only transcripts encoding an open reading frame resulted in a much reduced dataset, comprising 50,708 gall wasp transcripts corresponding to 24,916 genes and 178,522 oak transcripts corresponding to 53,529 genes (Table 1). Removing low-expression genes in DESeq2 (see Methods) retained 19,720 gall wasp genes and 42,926 oak genes in gall tissues, and 48,756 oak genes for normally developing oak bud tissues considered by each analysis. As we would expect given the increasing size of the gall wasp

**Table 1. Assembly metrics for the separated gall wasp (*Biorhiza pallida*) and oak bud plus gall tissue (*Quercus robur*) transcriptomes.**

| Assembly | Gall wasp–*B. pallida* | Oak—*Q. robur* |
|---|---|---|
| N50 * | 2338 | 1864 |
| Number of transcripts ** | 50708 | 178522 |
| Number of genes *** | 24916 | 53529 |
| Transcriptome span (Mb) | 78 | 232 |
| *BUSCO* complete (partial) % § | 97.7 (98.4) | 87.1 (98.0) |

* N50 is a weighted median contig length, such that 50% of the assembly is found in contigs of this length or greater.

** The total number of contigs produced by the *TRINITY* assembler.

*** The number of groups identified by *TRINITY* as representing separate loci.

§ *BUSCO* complete (and cumulative total including partially complete) genes after comparison to the Eukaryota database

**Table 2. Taxonomic assignments of gall dual-RNA-Seq reads, by stage and replicate (1–12), expressed as a % of the total mapped read set.** Full read counts and additional taxonomic sub-categorisations are provided in S1 and S2 Tables.

| Stage | Early | | | | Growth | | | | Mature | | | |
|---|---|---|---|---|---|---|---|---|---|---|---|---|
| Taxon of origin | 1 | 2 | 3 | 4 | 5 | 6 | 7 | 8 | 9 | 10 | 11 | 12 |
| *Q. robur* | 95.6 | 97.3 | 96.6 | 87.8 | 94.2 | 94.7 | 90.7 | 84.9 | 41.8 | 57.1 | 56.8 | 70.8 |
| *B. pallida* | 3.9 | 2.3 | 3.0 | 3.3 | 2.9 | 3.7 | 6.7 | 5.80 | 41.0 | 32.4 | 19.8 | 16.3 |
| Parasitoids | 1.0 | 0.2 | 0.2 | 0.7 | 2.5 | 0.7 | 1.0 | 7.1 | 10.1 | 7.1 | 13.6 | 3.7 |
| Fungi | 0.0 | 0.0 | 0.0 | 5.3 | 0.2 | 0.1 | 0.1 | 0.5 | 3.7 | 1.5 | 2.0 | 1.0 |
| Virus | 1.4 | 0.1 | 0.1 | 3.2 | 0.1 | 0.8 | 1.4 | 3.3 | 2.4 | 1.5 | 5.8 | 8.2 |

larvae, the proportion of gall wasp-derived reads in the entire gall transcriptome (the 'cecidome' [86]) rose from 2.3–4.0% in Early stage galls to 2.9–6.8% and 18–42% in Growth and Mature stages, respectively (Table 2). Oak and gall wasp transcripts comprised 78.6–99.6% of the total across the 12 gall replicates (Table 2). All galls yielded parasitoid-associated transcripts, with the lowest load in Early stage galls (Table 2, S2 Table). Viral and fungal sequences (considered further below) made up 0.1–8.2% and 0–5.3% of transcripts, respectively (Table 2).

## Gall wasp and oak genes show consistent patterns of stage-specific expression in *Biorhiza pallida* galls

Both gall wasp genes (Fig 3A) and oak genes (Fig 3B) in gall tissues showed largely consistent expression patterns across replicates within a given gall developmental stage. Gall stage accounted for 82% of variance in gall wasp gene expression and 50% of variance in oak gene expression. All gall wasp and oak genes showing differential expression between gall developmental stages at a $\log_2$-fold change of plus or minus one are listed in S8 and S9 Files, respectively. The numbers of gall wasp genes differentially expressed at more than our ±1-fold $\log_2$ difference threshold between Early and later stages increased during gall development, from 79 between Early and Growth stages to 371 between Early and Mature stages and 249 between Growth and Mature stages (Table 3). Differentially expressed gall wasp genes showed a strong bias towards higher expression in the younger gall stage in all between-stage comparisons (Table 3).

Numbers of oak genes differentially expressed between gall stages showed a somewhat contrasting pattern to gall wasp genes, with high numbers differentially expressed between Early stage galls and each of Growth (1,293) and Mature (1,890) stages, but only 98 between Growth and Mature stages (Table 3). Also in contrast to gall wasp genes, differentially expressed oak genes in gall tissues showed no consistent bias towards higher expression in younger gall tissues (Table 3).

## Oak gene expression in normally developing tissues shows partial differentiation between bud stages and strong differentiation between buds and leaves

Developmental stage explained 32% of the variance in oak gene expression in normally developing buds. In comparison to gall tissues, much lower relative numbers of transcripts from normal bud samples were identified as viral or fungal in origin (S3 Table). All oak genes showing differential expression between stages in normal bud development at a $\log_2$-fold change of plus or minus one are listed in S10 File. Only small numbers were differentially expressed between bud stages S0, S1, S3 and S4 (1–31 genes, depending on the stages compared;

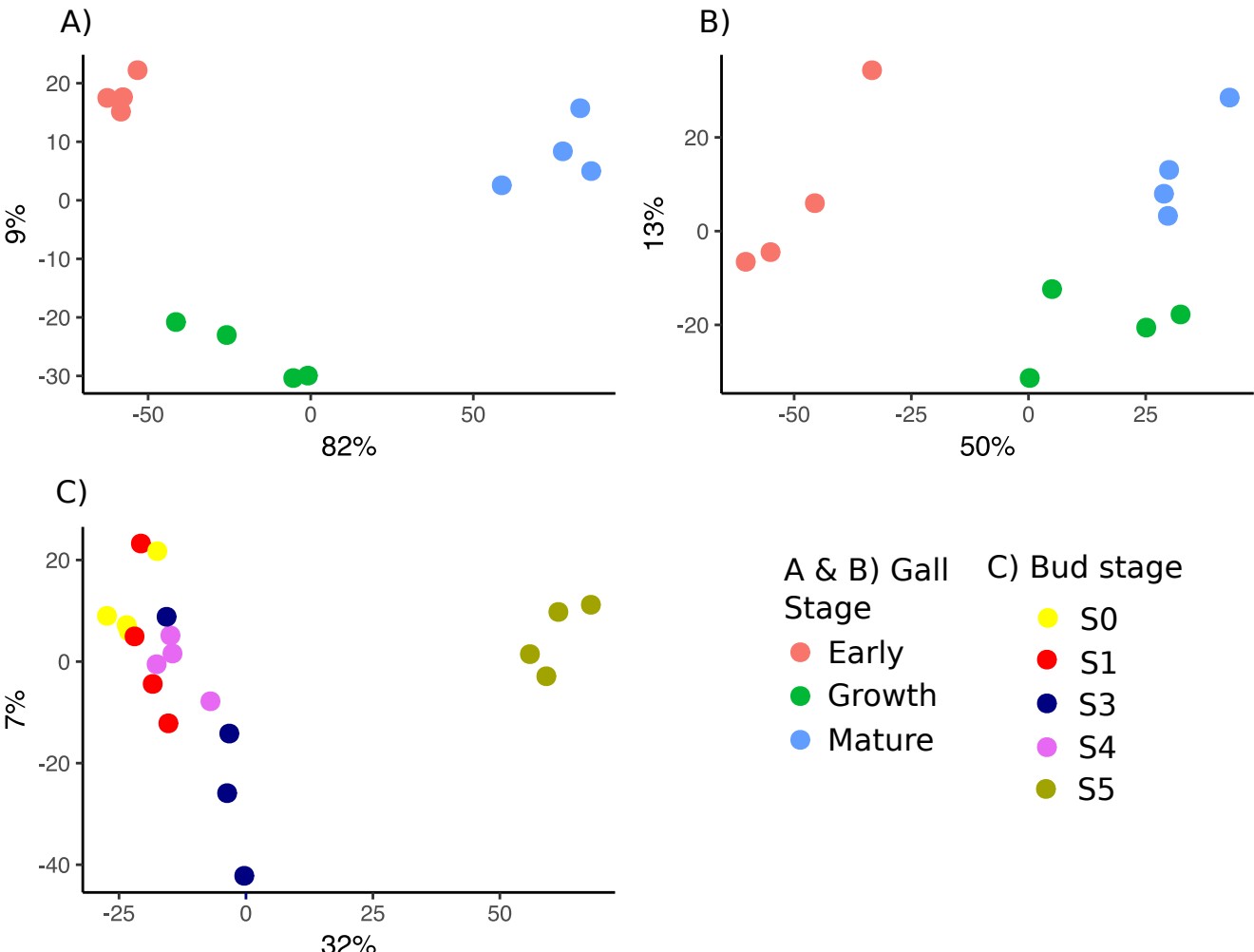

**Fig 3. Overall similarity in patterns of gene expression within and among developmental stages of *B. pallida* galls and normally developing oak buds.** Principle component analysis in *DESeq2* of (A) gall wasp genes in gall tissues, (B) oak genes in gall tissues, and (C) oak genes in normal (ungalled) bud tissues. X-axes correspond to principle component one and y-axes to principle component two, and axis labels give the percentage of variance these axes explain in each analysis. The colour legend for each stage is given in the bottom-right quadrant. For both gall wasp and oak genes in gall samples, biologically independent replicates for the same developmental stage cluster together. Growth and Mature stage tissues are more similar for host oak tissues than they are for larval gall wasp expression. Stages S0-S4 of normally developing buds show incomplete differentiation by stage, while mature leaf tissues (S5) are clearly distinct from all earlier stages. Lack of clear separation between stages S0-S4 is maintained if S5 replicates are removed from the analysis (S2 Fig).

Table 4), and gene expression patterns of these stages were incompletely separated (Fig 3C). In contrast, many genes (4,418) were differentially expressed between normal bud stage S4 and open leaves stage S5 (Table 4), and leaf tissues were clearly distinct from all earlier bud stages (Fig 3C). Lack of clear separation between bud stages S0-S4 was maintained if S5 replicates were removed from the analysis (S2 Fig). When data for bud stages S0-S4 were combined into a single bud developmental category (supported by lack of differential expression and overlapping PCA results, Fig 3C) the number of genes differentially expressed between bud stages S0-S4 and open leaves stage S5 rose to 4,981 (Table 4). In conclusion, identification of oak genes showing similar (or contrasting) trajectories of differential expression in gall and normal bud tissues thus primarily involves identifying those showing similar directions of differential expression between Early *versus* Growth or Mature gall stages, and between S4 buds and S5 leaves.

**Table 3. Counts of gall wasp and oak genes differentially expressed (DE) in gall tissues.** Host oak tissues are further summarised into the numbers of genes DE between gall stages that were not differentially expressed between any stages in normal bud development.

| Gall wasp (*B. pallida*) | Early *vs.* Growth | | Early *vs.* Mature | | Growth *vs.* Mature | |
|---|---|---|---|---|---|---|
| Direction of differential gene expression in earlier stage | Up in Early | Down in Early | Up in Early | Down in Early | Up in Growth | Down in Growth |
| Total number of DE* genes for this gall stage comparison | 75 | 4 | 283 | 88 | 210 | 39 |
| Oak (*Q. robur*) | Early *vs.* Growth | | Early *vs.* Mature | | Growth *vs.* Mature | |
| Direction of gene expression in earlier stage | Up in Early | Down in Early | Up in Early | Down in Early | Up in Growth | Down in Growth |
| Total number DE* genes in galls | 546 | 747 | 1045 | 845 | 63 | 35 |
| Number of DE genes that were not DE between any stages in normal bud development | 197 | 491 | 421 | 594 | 34 | 32 |
| % DE genes that were not DE between any stages in normal bud development | 36% | 66% | 40% | 70% | 54% | 91% |

* DE genes are defined as those showing at least a one-fold $\log_2$ change in *DESeq2* analyses. The Bud (S0-S4) vs. Leaf comparison represents results for combined developing bud stages (S0, S1, S3, and S4) *versus* mature leaf tissue S5. vs. = *versus*.

## Functional annotation of gall wasp and oak genes expressed during gall and normal bud development

Functional annotation of the gall wasp larval transcriptome was significantly poorer than for oak genes expressed in gall tissues, at 44% genes annotated *versus* 66% respectively. The subset of *B. pallida* genes differentially expressed between gall stages was more poorly annotated than the gall wasp transcriptome as a whole: functional annotations were identified for only 29% (22/75) of gall wasp genes more highly expressed in Early galls *versus* Growth and Mature stages, and none of the top five most highly expressed of these genes were annotated. Functional interpretation of changing gall wasp gene expression thus includes only a small proportion of the differentially expressed genes. In contrast, 89% (8225/9229) of differentially expressed oak genes for all contrasts in gall tissues and in normal buds were functionally annotated, allowing better-informed comparison of expression patterns in these two structures.

## Galled and ungalled buds show diverging patterns of oak gene expression during development

To compare galled and normal oak bud tissues, we identified oak genes and GO terms in gall and normal bud tissues showing similar or contrasting trajectories of expression during development of their respective tissues. A striking feature of our results is that most of the genes differentially expressed during gall development were not differentially expressed in normally developing bud tissues, and *vice versa*. The extent of overlap in differentially expressed oak genes declined during development of gall and normal bud development, compatible with increasing divergence between galler-induced and normal developmental trajectories.

Few genes were differentially expressed between normal bud stages S0–S4 (Table 3) and only five were differentially expressed in both gall and normal bud tissues (S4 Table). Four showed parallel expression trajectories of increasing expression through development of which two are unannotated (TRINITY_DN48733_c0_g1 and TRINITY_DN82248_c2_g1); one is a glyceraldehyde-3-phosphate dehydrogenase (TRINITY_DN55651_c1_g1) and one a cytochrome P450 (TRINITY_DN80279_c2_g1). Only one gene, a galacturonosyltransferase-like 10 (TRINITY_DN65887_c0_g1), showed increasing expression during gall development but decreasing expression during development of normal buds. Of the >4400 oak genes differentially expressed between normal bud stage S4 and fully opened leaves S5 (Table 4), 8.9–

**Table 4. Counts of gall wasp and oak genes differentially expressed (DE) in normally developing bud developmental stages.**

| Oak Buds (S0-S4) and Leaves (S5) | Bud S0 *vs*. Bud S1 | | Bud S1 *vs*. Bud S3 | | Bud S3 *vs*. Bud S4 | | Bud S4 *vs*. Leaf (S5) | | Bud (S0-S4) *vs*. Leaf (S5) | |
|---|---|---|---|---|---|---|---|---|---|---|
| Direction of differential gene expression | Up in S0 | Down in S0 | Up in S1 | Down in S1 | Up in S3 | Down in S3 | Up in S4 | Down in S4 | Up in S0-S4 | Down in S0-S4 |
| Total number of DE genes | 1 | 26 | 5 | 31 | 3 | 12 | 2108 | 2310 | 2000 | 2881 |

23.3% showed similar patterns of increasing or decreasing differential expression in gall tissues, depending on the direction of change and gall stages compared (Fig 4, S4 Table). Broadly similar numbers of shared differentially expressed genes were identified when expression in normal buds was compared with expression in either Early *versus* Growth or Early *versus* Mature stage galls (Fig 4). Much smaller numbers of oak genes (0.5–3.6% of those differentially expressed between S4 and S5 during normal bud development) showed contrasting trajectories of differential expression in normal bud and gall tissues. For example, only 10 of the 2108 genes (0.5%) that showed a significant increase in expression between normal bud stage S4 and fully open leaves S5 also showed a significant decrease in expression between Early and Mature stage galls. Shared differentially expressed genes comprised a higher percentage of all genes differentially expressed in gall tissues than normal oak buds (S4 Table).

To compare metabolism and development of gall and normal oak tissues, we identified oak GO terms with similar and contrasting trajectories of differential expression between gall developmental stages and between S4 bud and S5 leaf stages in normal bud development (Figs 4 and 5). Biological process (BP) and molecular function (MF) GO terms that were differentially expressed only during gall development are listed in S7 and S8 Tables, while those that showed similar patterns of differentially expression in both tissues (i.e. increasing in both or

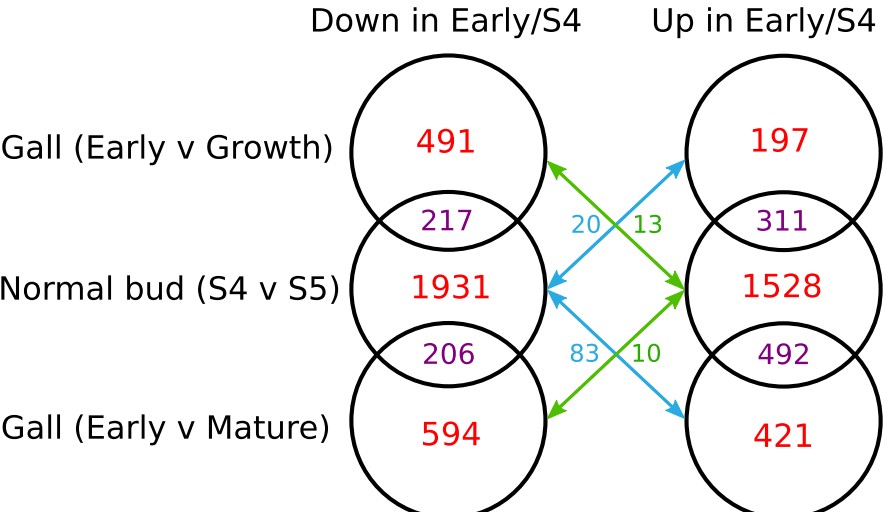

**Fig 4. Venn diagrams of numbers of oak genes that were differentially expressed in each, and both, of gall and normal bud tissues.** Venn diagrams are shown for genes that are up- or down-regulated in the earlier stage of each comparison as for Tables 3 and 4. Red numbers are the number of genes unique to that contrast, relative to ungalled buds for gall tissues, and relative to gall tissues for ungalled buds. Purple numbers show genes that are differentially expressed in the same direction in both galled and ungalled buds. For example, there are 311 genes that are upregulated in Early *versus* Growth stage galls and also in ungalled buds between S4 buds *versus* S5 leaves. Blue and green numbers are for genes with contrasting expression trajectories in gall tissues *versus* ungalled buds: blue numbers count genes that are upregulated in Early *versus* later stage galls but downregulated in S4 buds *versus* S5 leaves, while and green numbers count genes that are downregulated in Early stage galls but upregulated in ungalled S4 buds. These data are also given in S4 Table.

decreasing in both) are listed in S9 and S10 Tables. Biological process (BP) and molecular function (MF) GO terms that were differentially expressed during normal bud development are listed in S11 and S12 Tables.

*Biological process GO terms showing similar expression trajectories in gall tissues and normal buds.* Forty-seven biological process GO terms showed significantly higher expression in Early *versus* Growth stage galls and in normally developing S4 buds *versus* S5 leaves (S9 Table). These were grouped into 5 metabolic clusters by REVIGO treemap analysis (Fig 5B). As we might expect given rapid growth of both structures, GO terms elevated early in development of both galls and normal buds included many terms for cell division, cell cycle and proliferation (Fig 5B. 19/47 GO terms identified by orange circles) including DNA replication and endoreduplication/ regulation of DNA replication. Twenty-three biological GO term processes showed significantly higher expression in Growth *versus* Early stage galls and in normally developing S5 leaves *versus* S4 buds (S9 Table). These were grouped into seven metabolic clusters by REVIGO treemap analysis (Fig 5D), comprising processes associated with aging, catabolism, fruit dehiscence, lignin biosynthesis, secondary plant cell wall biogenesis, intercellular transport and response to nitrate. Fifteen of the same GO terms were shared with the 26 upregulated in Mature *versus* Early stage galls (S9 Table).

*Biological process GO terms showing contrasting expression trajectories in gall tissues and normal buds.* Fifteen biological process GO terms were upregulated in Early *versus* Growth stage galls, but not differentially expressed between any stages in normal bud development (S7 Table). These were grouped by REVIGO treemap analysis into 5 clusters of linked metabolic processes: response to brassinosteroid, leaf vascular tissue pattern formation, thymidine metabolism, cellulose catabolism and cutin biosynthesis (Fig 5A). Forty-five GO term processes were upregulated in Growth *versus* Early stage galls, but not differentially expressed between any stages in normal bud development (S7 Table). These were grouped into seven metabolic clusters by REVIGO treemap analysis (Fig 5C), comprising processes associated with carbohydrate metabolism, chorismate biosynthesis, fruit ripening, gibberellic acid homeostasis, L-glutamate transport, secondary plant cell wall biogenesis and response to Zinc ions. Fourteen of the same GO terms were shared with the 30 upregulated in Mature *versus* Early stage galls (S7 Table).

Our data thus suggest that galls of *Biorhiza pallida* share many metabolic and developmental characteristics with normally developing buds, but also that some genes show contrasting trajectories of differential expression between these tissues. Most genes that are differentially expressed between stages in gall development are also constitutively highly expressed in buds throughout development (S22 Table, mean expression of genes that are DE in galls but not buds for both experiments). In Early galls *versus* Growth this includes an endogenous endoglucanase and polygalacturonase (TRINITY_DN75537_c1_g1 and TRINITY_DN75938_c0_g1).

Further, many genes that are differentially expressed through gall development are not differentially expressed between any stages in normal leaf bud development. Similarity in oak gene expression patterns between gall tissues and normal buds declined as these tissues develop.

## Patterns in the expression of oak genes associated with alternative hypotheses for gall development

We compared expression patterns in galled and normally developing buds for specific candidate oak genes (S6 Table) relevant to alternative hypotheses of gall development.

i. *BCCP*. The 'galls as ectopic food storage organs' hypothesis predicts high expression of the acetyl CoA component *BCCP* during development of nutritive tissues in Early and Growth

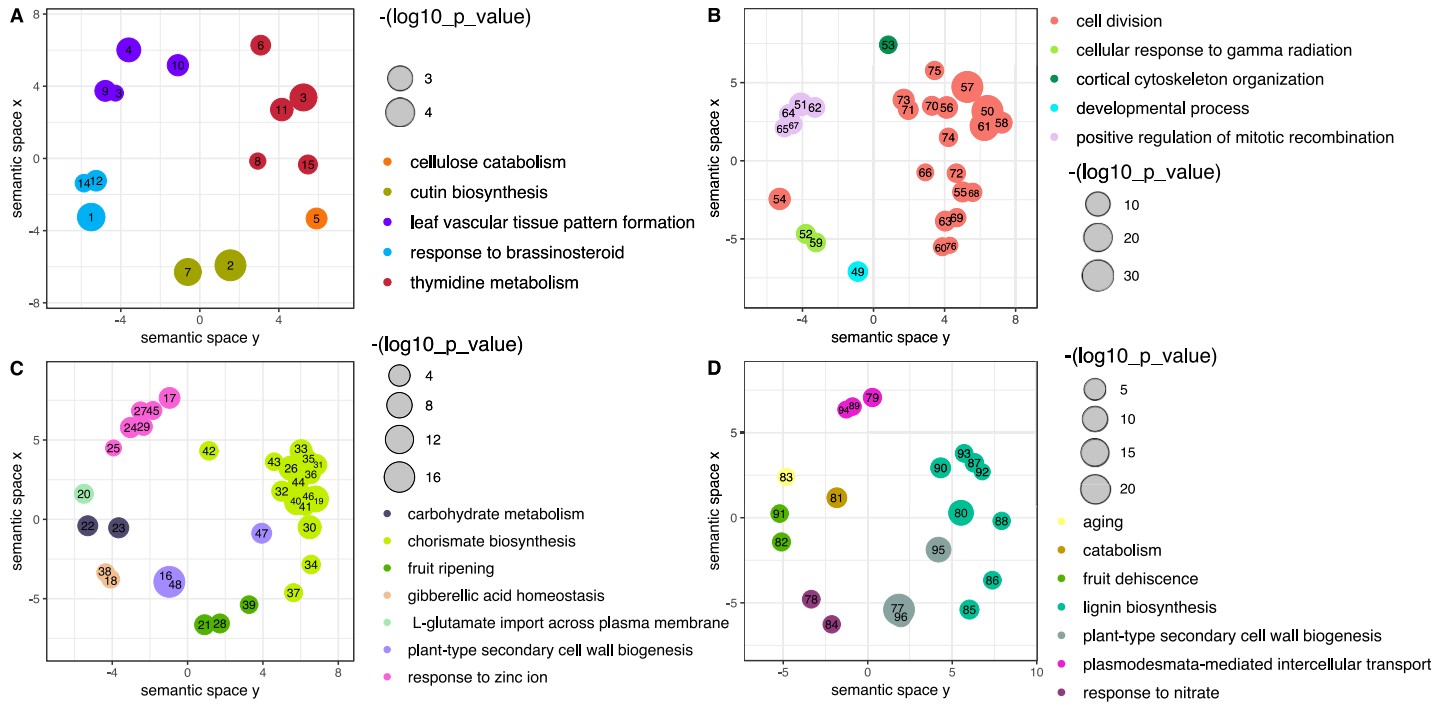

**Fig 5. REVIGO scatter plots of enriched biological process GO terms for oak genes expressed during development of galled and normal oak bud tissues.** A: GO terms that were upregulated only in Early *versus* Growth stage gall tissues, and not in comparisons between any normal (ungalled) bud stages. B: GO terms that were upregulated both in Early *versus* Growth stage gall tissues and in normal buds (S4) *versus* open leaves (S5). C: GO terms that were downregulated only in Early *versus* Growth stage gall tissues, and not in comparisons between any normal (ungalled) bud stages. D: GO terms that were downregulated both in Early *versus* Growth stage gall tissues and in normal buds (S4) *versus* open leaves (S5). Circle size is scaled by–$\log_{10}$ of the enrichment p-value with scales inset next to each plot. Groups of GO terms are coloured by cluster as classified by REVIGO treemaps (S3 Fig). There are fewer GO terms than for the equivalent *TopGO* comparisons as REVIGO reduced the redundancy in enriched GO terms. Numbers in the figures refer to the following GO terms and reference numbers: 1 = response to brassinosteroid (GO:0009741), 2 = cutin biosynthetic process (GO:0010143), 3 = thymidine metabolic process (GO:0046104), 4 = leaf vascular tissue pattern formation (GO:0010305), 5 = cellulose catabolic process (GO:0030245), 6 = asymmetric cell division (GO:0008356), 7 = wax biosynthetic process (GO:0010025), 8 = lignan biosynthetic process (GO:0009807), 9 = regulation of organ growth (GO:0046620), 10 = guard cell differentiation (GO:0010052), 11 = fatty acid metabolic process (GO:0006631), 12 = auxin-activated signalling pathway (GO:0009734), 13 = regulation of seed germination (GO:0010029), 14 = cytokinin-activated signalling pathway (GO:0009736), 15 = lipid catabolic process (GO:0016042), 16 = plant-type secondary cell wall biogenesis (GO:0009834), 17 = response to zinc ion (GO:0010043), 18 = gibberellic acid homeostasis (GO:0010336), 19 = chorismate biosynthetic process (GO:0009423), 20 = L-glutamate import across plasma membrane (GO:0098712), 21 = fruit ripening (GO:0009835), 22 = carbohydrate metabolic process (GO:0005975), 23 = proteolysis (GO:0006508), 24 = response to osmotic stress (GO:0006970), 25 = wound healing (GO:0042060), 26 = oxidation-reduction process (GO:0055114), 27 = response to herbicide (GO:0009635), 28 = regeneration (GO:0031099), 29 = response to biotic stimulus (GO:0009607), 30 = lignin biosynthetic process (GO:0009809), 31 = one-carbon metabolic process (GO:0006730), 32 = S-adenosylhomocysteine catabolic process (GO:0019510), 33 = shikimate metabolic process (GO:0019632), 34 = UDP-N-acetylglucosamine biosynthetic process (GO:0006048), 35 = ether metabolic process (GO:0018904), 36 = nitrate assimilation (GO:0042128), 37 = S-adenosylmethionine biosynthetic process (GO:0006556), 38 = sodium ion homeostasis (GO:0055078), 39 = mucilage biosynthetic process involved in seed coat development (GO:0048354), 40 = aromatic amino acid family biosynthetic process (GO:0009073), 41 = gluconeogenesis (GO:0006094), 42 = ureide catabolic process (GO:0010136), 43 = uracil catabolic process (GO:0006212), 44 = oxidative phosphorylation (GO:0006119), 45 = response to water deprivation (GO:0009414), 46 = beta-alanine biosynthetic process (GO:0019483), 47 = glucuronoxylan biosynthetic process (GO:0010417), 48 = cell wall organization (GO:0071555), 49 = developmental process (GO:0032502), 50 = cell division (GO:0051301), 51 = positive regulation of mitotic recombination (GO:0045951), 52 = cellular response to gamma radiation (GO:0071480), 53 = cortical cytoskeleton organization (GO:0030865), 54 = cell proliferation (GO:0008283), 55 = guard mother cell differentiation (GO:0010444), 56 = cytokinesis by cell plate formation (GO:0000911), 57 = microtubule-based movement (GO:0007018), 58 = microtubule-based process (GO:0007017), 59 = thigmotropism (GO:0009652), 60 = stomatal complex patterning (GO:0010375), 61 = cell cycle (GO:0007049), 62 = regulation of transcription, DNA-templated (GO:0006355), 63 = cotyledon development (GO:0048825), 64 = positive regulation of ubiquitin protein ligase activity (GO:1904668), 65 = positive regulation of anthocyanin metabolic process (GO:0031539), 66 = maintenance of floral organ identity (GO:0048497), 67 = positive regulation of cell proliferation (GO:0008284), 68 = epidermal cell fate specification (GO:0009957), 69 = endosperm development (GO:0009960), 70 = anastral spindle assembly involved in male meiosis (GO:0009971), 71 = DNA endoreduplication (GO:0042023), 72 = trichome branching (GO:0010091), 73 = mitotic spindle assembly checkpoint (GO:0007094), 74 = meiotic cell cycle (GO:0051321), 75 = phragmoplast microtubule organization (GO:0080175), 76 = polarity specification of adaxial/abaxial axis (GO:0009944), 77 = plant-type secondary cell wall biogenesis (GO:0009834), 78 = response to nitrate (GO:0010167), 79 = plasmodesmata-mediated intercellular transport (GO:0010497), 80 = lignin biosynthetic process (GO:0009809), 81 = catabolic process (GO:0009056), 82 = fruit dehiscence (GO:0010047), 83 = aging (GO:0007568), 84 = cellular response to salt stress (GO:0071472), 85 = S-adenosylmethionine biosynthetic process (GO:0006556), 86 = ethylene biosynthetic process (GO:0009693), 87 = regulation of salicylic acid metabolic process (GO:0010337), 88 = 3'-UTR-mediated mRNA destabilization (GO:0061158), 89 = nitrate transport (GO:0015706), 90 = oxidation-reduction process (GO:0055114), 91 = fruit ripening (GO:0009835), 92 = glycine catabolic process (GO:0006546), 93 = nitrate assimilation (GO:0042128), 94 = oligopeptide transport (GO:0006857), 95 = glucuronoxylan biosynthetic process (GO:0010417), 96 = cell wall organization (GO:0071555).

stage galls relative to Mature stage galls (in which nutritive tissues have been entirely consumed). In contrast to this prediction, we found no differential expression of *BCCP* across developmental stages of either gall tissues or normally developing buds, and two of five expressed copies were expressed highly in all gall and normal bud stages sampled (S6 Table).

ii. *ENOD* genes. The 'galls as modified somatic embryos' hypothesis predicts high expression of *ENOD* genes early in gall development. We found three genes coding for *ENOD* proteins containing phytocyanin domains to be significantly overexpressed in Early *versus* Growth stage galls. Of these, two were highly expressed in absolute terms (S6 Table), and were also expressed in normal bud tissues. One showed increasing expression during development of both gall tissues and normal buds (S4 to S5), while the second was not differentially expressed between any normal bud stages. These two proteins are present in eleven GO term gene groups that are enriched in Early *versus* later stage galls, including terms for cytokinesis, histone phosphorylation, anaphase, cell division and cell cycle. These data are compatible with a potential role for *ENOD* genes early in gall development.

iii. Oak chitinases. Chitinases can protect plant tissues against fungal attack, and are also involved in release of arabinogalactan signalling molecules from the phytocyanin-containing *ENOD* proteins considered above. The latter mechanism could be involved in gall development through action of one or both of gall wasp or oak chitinases. Numerous oak chitinases were expressed throughout development of galled and normally developing buds. The two most highly expressed chitinases were concordantly differentially and more highly expressed in mature galls (Mature > Growth and Early stages) and in older normal bud tissues (S5 > S4). Similar late stage host-plant chitinase expression was also observed in chestnut gall tissues attacked by another cynipid, *Dryocosmus kuriphilus* [87].

iv. Plant hormone-associated genes. Many auxin-related genes were expressed in *B. pallida* galls (n = 115) and in normally developing buds (118), and most (110) were expressed in both tissue types (S2–S4 Files and S8–S10 Files for annotations and significant genes respectively). Some genes expressed in galls and normal buds, including *Auxin responsive factor 5* and *Auxin responsive factor 9*, showed similar differential and higher expression in Early *versus* Growth galls, and in normal buds (S0-4) *versus* leaves S5. Oak calreticulins [88] were highly expressed in all stages of gall and normal bud tissues without differential overexpression. There was also no differential expression through development in gall or normal bud tissues of the jasmonic acid-amido synthetase (JAR1) gene, which is required by the jasmonic acid mediated signalling pathway, although the gene is expressed in all stages of bud and gall development. We detected no expression in gall or normal bud tissues of homologues of the key cytokinin transporter purine permease *PUP14* [89]. Ungalled Leaves (S5) and Growth stage galls both showed higher expression of the same aldehyde oxidase gene [90] than buds (S4) and Early stage galls, respectively.

However, several genes relevant to plant hormone signalling showed differential expression through development of galls but not normal buds. Early stage galls showed elevated expression of an auxin-related gene, *auxin-responsive protein IAA13* (a transcription factor involved in regulation of gene responses to auxin [91]), and a cytokinin-related gene, histidine kinase cytokinin receptor (*histidine kinase 5*). *Histidine kinase 5* fulfils a range of roles in *Arabidopsis*, including regulation of growth via the ethylene pathway [92], stomatal opening, and defence against pathogens [93]. Two *flavin-dependent monooxygenase* (*FMO*) [94] genes that were distinct from those differentially expressed during normal bud development were more highly expressed in Growth stage galls than in either Early or Mature stages. *FMO*s are involved in a

range of processes in plant metabolism including synthesis of auxin (IAA), processing of glucosinolates, and defence against pathogens [95].

## No demonstrated role for Symbionts and virus-like-particles in cynipid gall development

If symbiotic micro-organisms, viruses and/or virus-like particles (VLPs) play a fundamental role in cynipid gall induction, we expect to detect symbiont genomes and/or VLP coding sequence in assemblies generated from gall wasp genomic DNA libraries, and (with the exception of prokaryotes given our sequencing methodology) associated gall wasp larval gene expression.

We identified a supergroup A and B *Wolbachia* endosymbiont respectively in the *de novo* genome assemblies for *B. pallida* and the rose gall wasp *D. spinosa* (S13 Table). However, we found no bacterial or fungal symbiont genomes in any other gall inducing cynipids (or outgroup species; Table 5). These results, together with the patchy distribution of *Wolbachia* within and among gall wasp species [96,97], do not support a fundamental role for *Wolbachia*, other bacterial or fungal symbionts in cynipid gall induction. We also found no evidence for insertion of viral coat protein genes in the genomes of *B. pallida*, other gall wasps, or outgroup species, arguing against involvement of VLPs with recognisable homology to known viruses in transducing cynipid gall-inducing signals.

We sought transcripts that could derive from co-bionts in the dual RNA-seq data, noting that our RNA-seq methodology was not designed to detect expression of bacterial genes. Fungal gene expression was detected in all Growth and Mature stage galls of *B. pallida*, but was not detected in three of four Early stage gall replicates (Table 2), arguing against a role for fungal gene expression early in gall development. A small number of viral transcripts (782) were expressed at low levels in Early and Growth gall stages, and comprised a variable proportion of Mature gall transcripts reaching 8% in one sample (sample 12, Table 2). Viral sequences were identified as deriving principally from plant single stranded RNA viruses and derive from at least five different orders of viruses, with 'unidentified' the most prevalent grouping at the order level (S1 Table). However, no recognisably viral transcripts were expressed across all replicates of any gall developmental stage, providing no evidence as yet for viral symbionts in cynipid gall induction. The possibility of roles for currently unrecognised viral transcripts in cynipid gall induction cannot be excluded (see Discussion).

## Patterns of gall wasp gene expression during gall development

As noted above, functional annotation of the gall wasp transcriptome was relatively poor and functional interpretation of changing gall wasp gene expression through gall development is based on incomplete information. Notably however, gene products with predicted secretory peptide motifs and no transmembrane domain (44%) were abundant among differentially expressed genes including those otherwise un-annotated, a pattern that is compatible with their export from the gall wasp larva and roles in external manipulation and exploitation of the plant host. However, no overrepresented motifs other than transmembrane or signal peptide domains were found in Early genes *versus* the rest of the gall wasp gene set by MEME analysis [98], although we would not detect a motif through inter-stage comparisons if gall wasp virulence genes are necessarily constitutively expressed throughout gall-development.

Perhaps unsurprisingly given the low percentage of transcripts annotated, and the similar developmental state of the gall wasp larvae, Early *versus* Growth stage *B. pallida* transcripts were enriched for only two biological process and four molecular function GO terms (S14–S15 Tables). The four molecular function processes include pectin lyase (see below) and carbonate

**Table 5. Assembly metrics for gall inducing cynipid and parasitoid figitid wasp genomes.** Cynipidae and Figitidae are sister groups within the Cynipoidea that are estimated to have diverged 80 million years ago [55].

| Species and trophic group | N50 (bp) * | No. of contigs ** | Assembly size (Mb) | BUSCO complete (%) § | BUSCO partial (%) §§ | Assembly software |
|---|---|---|---|---|---|---|
| **Cynipidae** | | | | | | |
| *Aylax hypecoi* (gall inducer on *Hypecoum*, Papaveraceae) | 10635 | 119258 | 294 | 92 | 1 | SPAdes |
| *Diastrophus kincaidii* (gall inducer on *Rubus*) | 3366 | 125551 | 295 | 83 | 11 | SPAdes |
| *Diplolepis spinosa* (gall inducer on rose, *Rosa*) | 1747 | 1169600 | 511 | 86 | 8 | SPAdes |
| *Eschatocerus acaciae* (gall inducer on acacia) | 11180 | 25287 | 158 | 89 | 6 | MaSuRCA |
| *Pediaspis aceris* (gall inducer on sycamore, *Acer*) | 2703 | 389833 | 492 | 84 | 11 | SPAdes |
| *Synergus japonicus* (inquiline in oak cynipid galls) | 61843 | 12893 | 224 | 95 | 1 | MaSuRCA |
| **Figitidae** (parasitoids, sister group to the Cynipidae) | | | | | | |
| *Alloxysta arcuata* | 11631 | 76465 | 301 | 92 | 4 | MaSuRCA |
| *Parnips nigripes* | 2915 | 343555 | 470 | 86 | 10 | SPAdes |

* As defined in Table 1.

** Assemblies were filtered to remove all contigs less than 200 bp long.

§ The percentage of *BUSCO* core Eukaryota genes found in the assembly that are complete.

§§ The percentage of *BUSCO* core Eukaryota genes found in the assembly that are fragmented.

dehydrase activity. This increased to 38 biological process and 15 molecular function GO terms for Early *versus* Mature stage gall comparisons, including five chitin-associated terms for biological processes and chitinase and chitin-binding activity and pectin lyase activity for molecular functions (S14–S15 Tables). Few GO terms were enriched for genes more highly expressed in Mature *versus* earlier gall tissues stages (two and one biological process GO term and three and one molecular function GO term for Mature *versus* Early and Mature *versus* Growth stage comparisons, respectively); these included terms for digestion and nutrient reservoir activity, but there were no overrepresented motifs by MEME analysis [72].

As recorded in other herbivorous insects [99–102], we hypothesised that the genome of *Biorhiza pallida* would encode enzymes associated with metabolic breakdown of plant structural materials. In addition to roles in gall wasp nutrition, degradation products of cell wall components can act as plant signalling molecules [103,104] and so potentially play a part in gall development. We found fourteen plant cell wall degrading enzyme (PCWDE) loci coded within the gall wasp genome to be expressed through gall development, comprising six pectin/pectate lyases, four cellulases and four rhamnogalacturonan lyases (Table 6; S16 Table). Larvae from Early stage galls showed differentially greater expression of a pectin/pectate lyase *versus* both Growth and Mature stages, and three further pectin/pectate lyase genes were more highly expressed in Early *versus* Mature stages. None of the four gall wasp cellulases were differentially expressed across gall stages; two were highly expressed throughout larval development, while one (orthologous to other arthropod cellulases by blast annotation) was expressed only at low levels. The four rhamnogalacturonan lyase genes were also not differentially expressed across gall stages.

Early galls had elevated expression of seven gall wasp chitinases; two were highly expressed in absolute terms (S6 Table) and were differentially expressed in both Early *versus* Mature and Growth *versus* Mature comparisons. Of these seven, five contained a signal peptide and only

**Table 6. Plant cell wall degrading enzyme (PCWDE) genes in the nuclear genomes of cynipid gall wasps, their Figitidae sister group, and *Nasonia vitripennis* (Chalcidoidea).**

| Species | PCWDE type | | |
|---|---|---|---|
| | **Rhamnogalacturonan lyase** | **Pectin/Pectate lyase** | **Cellulase** |
| **Cynipidae** | | | |
| *Biorhiza pallida* | **Present**[*] | **Present** | **Present** |
| *Aylax hypecoi* | **Present** | **Present** | **Present** |
| *Diastrophus kincaidii* | **Present** | **Present** | **Present** |
| *Diplolepis spinosa* | **Present** | **Present** | **Present** |
| *Eschatocerus acaciae* | **Present** | **Present** | **Present** |
| *Pediaspis aceris* | **Present** | **Present** | *Absent* |
| *Synergus japonicus* | **Present** | **Present** | **Present** |
| **Other species** | | | |
| *Alloxysta arcuata* (Figitidae, Cynipoidea) | *Absent* | *Absent* | *Absent* |
| *Parnips nigripes* (Figitidae, Cynipoidea) | *Absent* | *Pseudogene* | *Pseudogene* |
| *Nasonia vitripennis* (Pteromalidae Chalcidoidea) | *Absent* | *Absent* | *Absent* |

[*] As the genomes are drafts it was not possible to accurately assign copy number for each gene in each genome. Counts of sequences for each enzyme type per species are given in S16 Table. Introns were predicted in all species and genes where PCWDEs were present except *D. spinosa* rhamnogalacturonan lyase and pectin lyase.

one encoded a signal peptide and transmembrane domain. All seven were also annotated with the GO term GO:0005576 for extracellular region. Taken together these are compatible with cellular export of these chitinases. Other gall wasp genes showing significantly elevated expression in Early stage galls (S6 Table) included a tyrosine-protein phosphatase, eleven carbonic anhydrases (involved in maintenance of pH balance and transport of carbon dioxide by hydration to bicarbonate [105]), venom acid phosphatase, and a glycine N-acyltransferase-like protein. We found no evidence for gall wasp production of plant hormone homologues in the differentially expressed genes, and neither did we detect expression of *BmIAO1* (or any close homologue), a gene involved in endogenous insect production of auxin [106]. We found no differential expression of putative xanthine dehydrogenase/aldehyde oxidase homologues for *BmIAO1* and only one expressed at low-level in this experiment (S6 Table). The Hessian fly, *Mayetiola destructor*, expresses genes that manipulate host plant hormones [107], but we found no homologues for these in the gene set for *B. pallida*. More generally, none of the *B. pallida* genes differentially expressed in Early galls were homologues of loci expressed by Hessian fly larvae during gall induction in wheat.

## Which gall wasp genes involved in gall development are novel compared to non-galling sister groups?

We searched the genomes of other gall-inducing and parasitoid cynipoids for orthologues of *B. pallida* genes identified as differentially expressed during gall development. Of the 75 protein-coding genes differentially expressed more highly in Early than Growth *B. pallida* galls (Table 3), 27 had orthologues in at least one other sampled cynipoid species (S17 Table). Of these 27, 14 (52%) had orthologues both in other gall-inducing cynipids and at least one parasitoid Figitid non-galling outgroup (*Alloxysta arcuata* and *Parnips nigripes*). One group incorporates all *B. pallida* Early stage differentially expressed carbonic anhydrase genes, and another incorporated venom acid phosphatases. These genes are not unique to gall-inducers as each orthogroup included orthologs from both galling cynipid and non-galling figitid genomes. Five orthologue groups were restricted to the cynipid gall-inducers: one encodes

pectate lyase PCWDEs, one a MBF2 transcription activator, and the remaining three are unannotated.

All three classes of plant cell wall degrading enzyme (PCWDE) found in *B. pallida* were also found in the nuclear genomes of other gall wasps, including an inquiline species (*Synergus japonicus*; Table 6; S16 Table) that induces its own nutritive tissues inside galls initiated by gall-inducing Cynipinae [54,108]. All of the PCWDE genes identified in *B. pallida* and other cynipoids show significant sequence similarity to PCWDE from plant pathogenic bacteria (S18 Table). However, three lines of evidence support the conclusion that these genes are within the insect genomes: (i) detection of transcripts in poly(A)-selected RNA-seq data is consistent with derivation from eukaryotic (rather than prokaryotic) transcripts; (ii) predicted PCWDE genes in gall wasp genomes include eukaryotic introns; and (iii) predicted PCWDE genes are flanked by one or more genes of unambiguously arthropod origin on the same contig (S19 Table). Orthologs to gall wasp PCWDE genes were not identified in nuclear genomes for the figitid parasitoids *P. nigripes* and *A. arcuata*, or the distantly related chalcid parasitoid *Nasonia vitripennis* (Table 6). However, for *P. nigripes* a BLAST search identified potential homologous sequence to *B. pallida* pectin lyase and cellulase. These are likely pseudogenes or fragments of a functional domain as, in contrast to all cynipid PCWDEs, the three putative loci identified all contain internal stop codons and were not predicted by *AUGUSTUS* [109] (Table 6). The PCWDE loci initially identified in *Biorhiza pallida* are thus almost or entirely restricted to the genomes of gall-inducing cynipids.

## Discussion

The galls induced on oak by *B. pallida* are an example of a common life history strategy amongst insects, mites, nematodes and other organisms. By manipulating the plant host to produce a protective structure that also supplies nutrients for growing larvae or sedentary adults, galling animals exploit and subvert plant developmental pathways to produce novel structures that benefit them, sometimes at measurable cost to their hosts. We have used genomic and transcriptomic analyses to explore the *B. pallida*–*Quercus robur* oak gall system to identify gene expression changes through gall development that are informative regarding the activity of the wasp within the gall and the response of the tree. Gall development involves expression of oak and gall wasp genes in repeatable, growth stage-specific patterns [26,86,87]. We compared oak gene expression in gall tissues and ungalled developing buds of *Q. robur* to identify genes and processes restricted to, and hence characteristic of, developing galls. To explore the evolutionary origins of genes involved in the galling phenotype we generated and analysed *de novo* draft genomes for a biologically diverse set of galling and non-galling Cynipoidea, looking for orthologues of candidate genes identified in *Biorhiza pallida*.

We sampled biologically independent replicates of three morphologically-defined gall stages, corresponding to Early, Growth, and Mature stage galls. Gall stage was a major driver of variance in the whole-gall RNA-seq data, suggesting that our sampling captured strong biological signal.

We used *de novo* genomes for the oak and gall wasp to separate the sampled whole gall transcriptome into oak, gall wasp and other co-biont compartments, and thus to assess, independently, the patterns of gene expression through gall development in each player. Oak transcripts dominated all gall stages, with the gall wasp component increasing as larvae developed. Gall wasp transcripts made up only ~3% of the Early gall transcriptome, but ~22% of the Mature gall. Fungal and viral infection of the galls was also detected and variable between replicates. This was in contrast to normally developing buds, which had negligible viral and fungal infection levels. Analyses of the patterns of oak and gall wasp gene expression accounted for

these co-bionts by controlling for their effect during differential expression testing (see Methods).

We found marked divergence in oak gene expression between galled and ungalled host tissues, as observed in cynipid-induced chestnut galls and phylloxera-induced galls on grape vines [25,87]. The clear developmental trajectory of gall tissues also contrasted strongly with the pattern observed in normally developing ungalled buds, which showed little transcriptomic differentiation between recognised morphological stages through budburst, but clear differentiation between opening buds and fully open leaves.

We were able to functionally annotate most oak transcripts based on the large body of data and analysis available for model and crop plants. However, the gall wasp transcripts were relatively poorly functionally annotated, with many fewer sequence similarity matches and known domains. Functional annotation was particularly scarce for gall wasp genes that were differentially expressed between Early gall and later developmental stages. The annotational novelty in the gall wasp larval transcriptome parallels that for gall wasp ovaries and venom glands, which represent possible sources of maternal stimuli involved in the initiation stage of gall induction [49]. Around 90% of differentially expressed venom and ovary-specific transcripts are novel in *B. pallida* and the rose gall wasp *Diplolepis rosae* while equivalently expressed genes are better annotated. This suggests that lack of annotation for differentially expressed genes reflects their novelty rather than a general lack of annotation for gall wasp genomes [49]. Similarly low levels of annotation and high annotational novelty have been reported in the immature stages of other gall inducers, including fig wasps and Hessian fly [8,21,22]. The long divergence of gall wasps from non-galling ancestors [55] coupled with potentially rapid (co-)evolution of genes underlying the galling interaction could well leave little signal of orthology between contemporary lineages. We now return to the four questions originally posed in the Introduction.

## What, in plant terms, is a cynipid gall?

Different galling organisms induce galls on different plant tissues, including meristematic tissues and developing organs of both aerial and root systems. These different targets, and the different galls induced, may result from both shared and taxon-specific manipulation of core plant processes. Cynipid galls have morphologies that are specific to the inducing wasp species and generation, and thus represent a specific response of the host to the particular infection experienced. *Biorhiza pallida* sexual generation galls are derived from oak bud meristematic tissues which, in normal development, would generate leaves or flower parts, but (in contrast to some other insect-induced galls [25]) the galls do not obviously resemble these plant organs. Two related hypotheses for the oak organ similarities of cynipid galls have been proposed previously from anatomical and developmental studies [1,59,60]: 'galls as ectopic food storage organs' and 'galls as modified somatic embryos'. We used dual RNA-seq expression data to identify genetic systems expressed in the oak tissues, and compared these to known plant organ systems to assess expression phenotype resemblances. Overall, we found gene expression and inferred developmental process in galled and ungalled buds to diverge substantially through development, and more support for 'galls as modified somatic embryos' than 'galls as ectopic storage organs'.

Developing buds and Early stage gall oak tissues share many processes of cell division, cell cycle, and chromosomal organisation (Fig 5) that are fundamental to organogenesis in plants. As galled and ungalled tissues develop, their trajectories of differential oak gene expression diverge, to the point that genes differentially expressed in Mature galls relative to earlier stages show little or no overlap to genes differentially expressed in leaves relative to earlier stages in normal bud development. While the sexual generation gall of *Biorhiza pallida* is clearly very

different from the leaf into which the host bud would otherwise develop, more in depth analysis of gene expression is required to query whether cynipid gall development involves pathways involved in normal plant organogenesis, such as the flower development pathways recently demonstrated in phylloxera galls [25].

Some differential oak gene expression between gall developmental stages can be related directly to observed developmental or homeostatic processes in the galls. For example, differentially high expression of genes involved in cell wall growth and vascularization, such as *Auxin responsive factor 5* in Early stage galls, and of other genes later in gall development, matches observed development of outer gall tissues and of vascular bundles supplying the larval chambers. Given that cynipid galls expand rapidly during the Growth stage, the absence of elevated expression of cell division-associated genes in Growth *versus* Early stage galls is perhaps surprising. However, anatomical studies show that much of the growth of oak cynipid galls is accomplished by increase in cell volume (hypertrophy) rather than by cell division [37,46]. This parallels the similar transition from cell division to cell expansion observed as normally developing buds flush to fully developed leaves [110–112].

Both developing galls and ungalled buds expressed oak gene systems associated with cell division and nuclear endoreduplication. In terms of oak cynipid gall development these findings are consistent with previous *in situ* hybridization analysis of gall development in *B. pallida*, which suggested that endoreduplication and somatic embryogenesis are primarily associated with nutritive tissues immediately surrounding the larva, while high rates of cell division are associated with rapid expansion of the outer parenchyma [59]. That these patterns may be common in cynipids is suggested by transcriptomic evidence from galls induced in chestnut by another gall wasp, *Dryocosmus kuriphilus* [113]. Transcriptomic sampling of oak genes in specific gall tissue compartments, rather than the entire gall, is required to test these ideas.

The 'galls as ectopic food storage organs' hypothesis was based on detection in earlier metabolomic studies of *BCCP* expression by gall tissues, associated with seed development in *Arabidopsis* and food storage organ development in other plants [59,60,70]. We found *BCCP* expression throughout bud development into mature leaves, and across all three stages of gall development, confirming previous work [59,60]. While our data do not rule out a role for *BCCP* expression in nutrient processing in cynipid galls as originally hypothesised, they confirm a more general role for products of these genes during both gall and normal bud and leaf development.

Plant somatic embryogenesis is characterized by expression of phytocyanin domain-containing arabinogalactan proteins (AGPs), proteoglycans that carry sets of arabinosyl and galactosyl residues. AGPs are naturally cleaved by action of plant chitinase enzymes and then act as signal molecules [114–118]. These signalling-associated AGPs are present in a range of plant tissues, including seeds [119,120] and we found them to be highly expressed in all normally developing bud stages (S0-S4) except leaves (S5). Signalling AGPs are encoded by a subset of the *ENOD* genes [121,122] involved in legume root nodule development [123,124]). We observed upregulation of *ENOD* AGPs in Early stage galls, and suggest that the somatic embryogenesis-like state observed in young cynipid galls may be specifically induced by signals derived from oak AGPs by action of gall wasp chitinases (see below).

Proteomic work on leaf galls induced by other oak cynipids (*Cynips* and *Neuroterus* species) identified overproduction of the enzyme S-adenosyl methionine (SAM) synthase in gall tissues [105]. We found four genes encoding the same enzyme to be differentially over-expressed in Growth stage relative to Early stage galls, two of which are only differentially expressed in gall tissues (TRINITY_DN55180_c4_g1 and TRINITY_DN69171_c2_g1). SAM synthase catalyses the production of SAM from methionine, and the SAM in turn is a substrate for the synthesis

of a range of important plant metabolites, including ethylene and polyamines. Possible roles for these metabolites discussed in [105] include a range of processes relevant to gall induction that includes meristem development [125], regulation of *Rhizobium*-induced root nodulation in legumes [126], organogenesis and fruit maturation [127]. Further work is required to understand the role(s) of these genes in gall tissues.

## What organism induces the *B. pallida* oak gall?

We found no support for symbiont involvement in *B. pallida* gall development. Although gall development is known to require a living gall wasp larva, it has remained possible that the inducing organism is not the wasp but rather a symbiont. Symbiotic bacteria such as *Wolbachia* are present in many galling insects, and have been shown to influence the development of plants attacked by leaf miners [73]. Bacterial endosymbionts are abundant in the guts of both larval and adult Hessian flies, though their role in the Hessian fly-wheat interaction is unknown [74]. While our poly(A)-selected transcriptomic analysis was not intended to quantify bacterial gene expression, our analyses of adult gall wasp holobiont genomic data found no consistent association between gall wasps and specific bacterial endosymbionts. We identified *Wolbachia* in genome sequence data from *B. pallida* (a supergroup A strain) and the rose gall wasp *Diplolepis spinosa* (a supergroup B strain), but not in the other gall wasp or Figitidae sister group genomes analysed. *Wolbachia* prevalence and incidence vary substantially within and among gall wasp species, suggesting multiple gains and losses of infection [96,97] and arguing against a fundamental role for *Wolbachia* in cynipid gall development. We detected transcripts from plant viruses and plant-associated fungi in gall tissues, but neither comprised taxa that were consistently detected in cynipid genomic libraries, nor showed expression patterns across gall stages consistent with a causal role in gall development. However, new viral taxa continue to be discovered [128,129] and it remains possible that unrecognised viral symbionts contribute to the large numbers of differentially expressed but unannotated genes attributed to the gall wasp in our experiment. A role for viruses (or any other currently unrecognisable symbiont) in gall induction thus cannot be wholly excluded. However, our data are consistent with phylogenetic patterns in gall traits that support a major role of gall wasp nuclear genes in gall development.

## What processes are involved in gall wasp manipulation of plant development?

Our data are compatible with secretion of effectors from the gallwasp into surrounding host plant tissues, we hypothesise from the enlarged larval salivary glands. We find no evidence for effector delivery via virus-like particles (VLPs).

(i) *How does the gall wasp larva export stimuli to the plant*? In other plant pathogenic animals, including nematodes, gall-inducing fig wasps and Hessian flies, highly expressed novel proteins with secretory signal peptides have been proposed to represent putative effectors [21,46,130]. Most Early stage gall wasp transcripts had no informative functional annotation, but many highly-expressed loci encoded proteins with secretory signal peptides. Of the 75 genes more highly expressed in Early stage gall wasp larvae, 44% (33) had signal peptides and no transmembrane helix, compatible with secretion of these proteins into surrounding host tissues. These are obvious candidates for further study. Whether these proteins are secreted or excreted by the wasp is unknown, but they could include novel effectors. Their general lack of informative annotation, and the absence of any conserved motifs among them, will make analysis challenging. A striking feature of the unannotated but highly expressed genes of Early stage gall wasp larvae is their lack of homology with similarly unannotated protein coding

genes expressed during gall induction and development by Hessian flies or any known genes (via comparison to the non-redundant protein database). This lack of homology suggests that even if these gall inducers are targeting similar developmental pathways in their plant hosts, they are doing so in taxon-specific way. This lack of homology is also concordant with the lineage-specificity of genes involved in plant manipulation by the aphid *Myzus persicae* relative to other arthropods [131].

Parasitoid wasps deliver DNA and protein to their hosts through virus-like particles, produced from viral genes incorporated into the wasp nuclear genome [79,81]. It is possible that a similar system could be used to deliver gall inducing stimuli in cynipid galls. However, we found no evidence of Early gall expression of gall wasp homologues of genes coding for viral coat proteins, arguing against export of gall inducing stimuli in virus-like particles with homology to any documented virus [128]. Cambier et al [49] also found no evidence for VLPs in the ovaries and venom gland transcriptomes of *B. pallida* and a rose gall wasp, *D. rosae*. The possibility that gallwasps use VLPs derived from currently unrecognised viral taxa cannot, however, be excluded.

(ii) *What signals might be involved*? Plant development is regulated by networks of peptide and other hormones, cell interactions and cell-autonomous processes. Gall wasps could drive gall development by intervening at key steps in normal plant developmental processes, initiating pathways that then autonomously generate structures that support gall wasp development. In a psyllid leaf gall system, the host gall response included transcriptional upregulation of several loci annotated as involved in responses to auxin, but the biological annotation of the insect genes was not reported [132]. Auxin (indoleacetic acid, *IAA*) and cytokinins have been extracted from the bodies of a range of insect gall inducers including aphids, sawflies, gall-midges and cynipids [36,94,132–138], and while the source (insect, symbiont, host plant) of these effectors has in some cases yet to be determined beyond doubt, ability to synthesise *IAA* may be widespread in insects [106]. It has been proposed that gall inducing insects acquired the necessary plant synthetic enzymes through microbial symbiosis or lateral gene transfer [139,140]. Indeed, it has been proposed that plants themselves acquired IAA synthesis from bacteria [141]. We found no evidence for Early gall stage upregulation of gall wasp transcripts with recognisable homology to plant genes involved in auxin- or cytokinin-related processes, including the recently discovered *BmIAO1* gene [106]. We also found no sequences with recognisable affinity to cytokinin-related plant genes in cynipid genomes. We found no evidence for consistent gall wasp association with a bacterial or fungal endosymbiont that could produce plant hormone homologues or analogues. There is thus currently no evidence to support the hypothesis of horizontal transfer of genes regulating plant hormonal systems into the gall wasp genome.

This does not mean that auxins and other phytohormones are not involved in cynipid gall development, rather their involvement is likely to be downstream of the key gall inductive and maintenance signals. The oak *Auxin responsive factor 5*-like gene, highly expressed in Early stage galls, is typically associated with auxin-induced roles in axis formation and/or the development of gall xylem and phloem tissues [142,143]. It is also possible that gall wasps interfere with, or produce, phytohormones using enzymes that have no informative sequence similarity to the plant genes that normally regulate and produce these messengers. Direct assays of phytohormones or their precursors in the bodies of gall inducers, and evidence of endogenous production rather than concentration of plant or symbiont products [94,106,134,138,144], are more powerful tools in assessing the roles and sources of auxins, cytokinins and other plant hormones.

## Which gall wasp genes involved in gall development are novel compared to non-galling sister groups?

(i) *Gall wasps produce their own plant cell wall degrading enzymes (PCWDEs)*. To exploit gall tissues, the developing wasp larvae must access and digest plant tissues. Digestion of the complex plant cell wall is accomplished by gut microbiota in most insect herbivores, but a growing number of studies have detected endogenous PCWDE genes in insects [145–148]. These genes are derived from a combination of horizontal gene transfer events from bacteria (HGT) [149,150] and expansion of PCWDE gene families within insect genomes. Three classes of gall wasp-encoded PCWDEs were expressed during cynipid gall development: pectin lyases, rhamnogalacturonan lyases and cellulases. Gall wasp pectin lyase activity (GO:0047490) was differentially overexpressed in Early relative to later stage galls, and two distinct gall wasp cellulase genes were highly expressed throughout gall development. PCWDE genes are also expressed in the venom glands of *B. pallida* [49], as is a venom acid phosphatase.

Though the genomic origin of venom gland-expressed genes was not demonstrated, two were hypothesized to have been incorporated in the gall wasp genome following horizontal gene transfer from a bacterial origin [49]. Our genomic analyses confirm that the cellulases expressed in adult and larval gall wasps are encoded by gall wasp genes, through detection of (a) spliceosomal introns indicative of eukaryotic genome architecture, (b) sequence contiguous with the cellulases that contains genes of obvious insect origin in multiple gall wasp species, and (c) polyadenylated (i.e. non-bacterial) cellulase mRNAs in the *B. pallida* transcriptome. As hypothesised for the *B. pallida* venom gland cellulases, larvally-expressed cellulases are most similar to either bacterial sequences or to genes in distantly related insect herbivores for which bacterial origins have been proposed [100]. Interestingly, *BLAST* comparison shows that different and divergent cellulase genes are expressed in adult *B. pallida* venom glands and in larvae, aligning to different contigs in the *B. pallida genome* (S21 Table). This could reflect adaptation to differing roles and/or host plant targets, such as adaptation of specific gene copies for high and tissue-specific expression via a secretory organ-specific promoter. This is hypothesised to originate from the venom gland in the adult female [49] and is hypothesised to be the salivary glands in the larva [43,56]. It may also be attributable in part to differing methodologies of transcriptome sequencing and assembly between this study and [49].

Gall wasp PCWDEs could be involved in larval hatching and maternal venom gland PCWDEs could contribute to formation of the initial chamber adjacent to the egg into which the hatching larva moves [40]. Larval PCWDEs probably function to break down the walls of nutritive cells in the larval chamber [2,151,152], and could also facilitate passage of gall induction effector proteins or maintenance factors (potentially including cell wall breakdown products themselves) into host plant tissues by making cell walls permeable to the passage of signal macromolecules. Intracellularly-acting effectors have been identified in other galling insects including aphids and hessian flies [23,153–155]. Both have mouthparts capable of penetrating plant cells to aid effector delivery, and this process could be assisted enzymatically in gall wasps. Bacterial expression assays of gall wasp PCWDEs are required to determine their substrate relationships and identify their roles *in vivo*.

(ii) *A PCWDE repertoire may be a synapomorphy of gall-inducing and inquiline cynipids.* Orthologous genes encoding all three classes of *B. pallida* PCWDEs were found in the genomes of other cynipid wasps, including the inquiline *S. japonicus* and the gall-inducing species *A. hypecoi* and *E. acaciae* (Table 6), representing members of three deeply diverging tribal lineages within the Cynipidae [54]. The same genes were absent from (or detected only as pseudogenes in) the genomes of non-galling Figitid wasps, the sister group of gall wasps

whose parasitoid life history is thought to represent that ancestral state from which cynipid gall induction is derived [39]. This suggests that a multilocus repertoire of PCWDEs is a derived trait that evolved in the shared common ancestor of gall inducing and inquiline cynipids, i.e., a synapomorphy. Gall wasps evolved from a parasitoid of insect hosts concealed within plant tissues [55], and shifted to phytophagy either prior to or concomitantly with the evolution of gall induction [54,55]. Acquisition of PCWDEs by the common ancestor of gall wasps may have helped larvae migrate through plant tissues in search of insect hosts (as a parasitoid), feeding sites (as a herbivore) or assisted in penetration of plant tissues by the adult female ovipositor prior to the evolution of gall-induction. Wider sampling of cynipids and their relatives in the Cynipoidea is required to reconstruct the origin(s) and diversification of the cynipid PCWDE repertoire.

## Chitinases, arabino-galactan glycoproteins and a model for cynipid gall induction

Gall wasp chitinases are a notable exception to the general paucity of functional annotation of cynipid genes differentially overexpressed in Early and Growth stage galls. Homologues of these chitinases were present in other gall wasps and our sampled outgroups. We hypothesise that Early stage over-expression of gall wasp chitinases may play a role in gall formation through interaction with oak *ENOD* gene products.

Two oak *ENOD* (early nodulin) arabinogalactan protein (AGP) transcripts were upregulated in Early stage galls. *ENOD* genes were initially identified for their roles in *Rhizobium* nodule development in legumes [156], but are widely present in vascular plants [157,158]. Legume *ENOD* genes are specifically induced by nod factors during the earliest stage of *Rhizobium* nodule development, and have been co-opted for nutritive benefit by reniform nematode pathogens of soybeans [159]. *ENOD* genes could potentially offer a toolkit for host manipulation by parasites more widely through their key roles in plant developmental pathways [159]. The upregulated oak *ENOD* AGPs in *B. pallida* galls belong to a group containing a phytocyanin-like domain that have been implicated in somatic embryogenesis in plants [116]. AGPs carry glycan side chains containing glucosamine and N-acetyl-D-glucosaminyl residues, which can be cleaved from the molecule by chitinases [114,115]. Action of plant chitinases on AGPs promotes somatic embryogenesis, implying that the released glycan moieties may have a role in signalling [114,160]. We hypothesise that the wasp chitinases highly expressed in Early stage galls could manipulate oak developmental pathways by similarly cleaving *ENOD* AGP proteins, thus inducing somatic embryogenesis.

All of the cynipid genomes we examined contained at least one chitinase gene. These chitinases were most similar to the GH18 family of insect chitinases, exemplified by the *N. vitripennis* chitotriosidase-1-like protein. GH18 chitinases function in turnover of extracellular chitin-containing matrices, such as the insect cuticle [161] during larval and pupal moulting [162]. However, we suggest that chitinase expression in larval *B. pallida* is probably not associated with moulting because larvae grow very slowly in Early stage galls (when chitinase expression is high) and then grow rapidly and moult in Growth and Mature galls (when expression of the same chitinases is low in our data).

Additional possible roles for insect chitinases are breakdown of chitin in the eggshell, protection against attack by fungi, and suppression of plant defence responses to large chitin molecules [163–166]. Most insect egg shells do not contain chitin [167,168], though where chitin is present newly-hatched larvae do show chitinase activity [169]. However, young larval *B. pallida* gall wasps are separated from their eggshells by their own movement into plant tissues, and by the development of plant gall tissues around them [1]. A role in eggshell degradation for early chitinase expression is thus undemonstrated. Gall wasp chitinases could potentially

play a role in protection of the gall wasp and gall tissues from fungal attack, given that fungal transcripts were detected in gall samples. However, chitinase expression in Early stage galls was not correlated with fungal infection of galls across replicates (S2 Table), and we thus consider it unlikely that the observed chitinase production was part of an induced anti-fungal defence. It is also interesting that larval chitinases are distinct from the chitinase expressed in the adult female venom gland (S21 Table), which is probably injected along with the egg during oviposition and for which an anti-fungal role has been suggested [49]. If our hypothesis is correct, we expect chitinases that interact with the host to be exported from the gall wasp larva. In Hessian flies, chitinases are expressed by the salivary glands, making them potential effectors in that system [21].

We speculate that secreted gall wasp chitinases act in Early stage galls to mimic endogenous plant chitinases. By cleaving glucosamine and N-acetyl-D-glucosaminyl residues from oak AGPs in the extracellular matrix they could drive induction of somatic embryogenesis-like dedifferentiation and cell division in host tissues. To test this model, future work could identify the larval tissue of origin of the chitinases, localise the gall wasp chitinases in the developing gall, define their enzymatic activity on oak glycoproteins *in vitro*, and explore expression patterns of orthologues in other galling species.

## Conclusions

We have identified candidate genes associated with gall development in both the gall inducer and the host plant in the *B. pallida*-oak system. Functional inference from oak gene expression, most of which was annotated, correlates with observed development of gall phenotypes. In contrast, most gall wasp genes differentially expressed at the key early post-initiation stage had no informative functional annotation. However, the fact that many are predicted to be secreted suggests that they could include novel parasitic effectors of plant development. We hypothesise that high expression of host arabinogalactan proteins and of gall wasp chitinases in Early stage galls interacts to generate a somatic embryogenesis-like process in nutritive tissues surrounding the gall wasp larvae. This mechanistic framework provides testable hypotheses for future functional dissection of cynipid gall development.

## Methods

### Sample collection, RNA extraction and sequencing

We sampled sexual generation galls of *Biorhiza pallida* near Blandford Forum, Dorset in 2011 and sampled normally developing buds at Dalkeith Country Park in 2018, Midlothian, United Kingdom. All tissues were sampled from oaks morphologically identifiable as *Quercus robur*. Four biological replicates, each from a separate tree, were collected for each of the following three gall developmental stages, giving 12 samples in total. (i) Early stage galls (Fig 1A and 1B. Gall diameter <5mm, often fully or partly concealed within bud scales; for illustration of all 4 biological replicates, see S1 Fig). (ii) Growth stage galls (Fig 1A, 1D and 1E. Gall diameter 20-30mm, advanced larval chamber and nutritive tissue development, larval length <1mm). (iii) Mature galls (Fig 1A, 1F and 1G; epidermis brown, with a papery texture, internal tissues lignified, larvae active and ≥3mm long). Four biological replicates, again each from a separate tree, were collected for each of bud tissue developmental stages S0-S5 (Fig 1A). All galls were rapidly sliced into 1mm thick sections, bud tissues diced into small pieces, and a cross section of leaf tissue from the centre of the leaf perpendicular to the petiole of approximately 2 cm width was sliced and immediately immersed in RNAlater (Ambion) in the field. RNA extractions were made for each replicate using the RNEasy plant mini kit extraction protocol (Qiagen) and stored at -80˚C until sequencing.

Presence of gall wasp mRNA in extractions was confirmed using reverse transcriptase polymerase chain reaction (RT-PCR) amplification of a pair of exon-primed intron-crossing (EPIC) gall wasp loci, *Receptor for Activated C Kinase 1 (RACK1)* and *Ribosomal Protein L37 Rpl37* ([170], further refined for cynipids by James Nicholls (personal communication). A *B. pallida* genomic DNA positive control was used to indicate cDNA amplification over genomic DNA carry-over.

Sample quality was assessed for RNA purity by 260/280 and 260/230 ratios measured on a NanoDrop spectrophotometer (Thermo Scientific), followed by RNA integrity analysis by Agilent 2100 Bioanalyzer (Agilent Technologies) total RNA nano trace. No samples showed visible degradation on Bioanalyzer traces. Gall and ungalled bud/leaf samples were prepared as 100 and 150 base pair paired-end TruSeq libraries respectively by the NERC Edinburgh Genomics facility. Gall and ungalled bud libraries were multiplexed and sequenced separately on one lane of an Illumina Hi-Seq sequencer. The eight Early and Growth stage gall libraries were sequenced on an additional lane to increase numbers of gall wasp reads sequenced, since in these stages gall wasp RNA comprises only a small fraction of whole gall RNA.

## Transcriptome assembly, quantification and taxonomic assignment

Reads (ENA PRJEB13357; S20 Table) were quality and adapter trimmed with default parameters in *FASTP* version 0.19.3 [171] and results checked in *FASTQC* version 0.11.7 [172]. All reads were then assembled in *TRINITY* version 2.6.6 [173]. Transcripts were quantified in *SALMON* [174] with "—seqBias", "—gcBias", and "—rangeFactorizationBins 4" options and converted to gene-level counts using the Bioconductor package *TXIMPORT* [175] (Rscript: S1 File). Transcripts were annotated in the *TRINOTATE* pipeline [176] (S2–S4 Files). To assign taxonomic origins a *TRINITY* SuperTranscript [177] was created from each group of associated transcripts representing a gene and compared against a combined reference using *MINIMAP2* in 'splice' mode [178]. This reference consisted of the *B. pallida* genome sequence [50], the *Q. robur* haploid genome sequence (PM1N) [179] and four chalcid parasitoid genomes [51]. The four chalcid parasitoids are common gall inhabitants and were used to remove parasitoid hymenopteran sequences that were not gall wasp-derived. They were *Cecidostiba fungosa* (Pteromalidae), *Megastigmus dorsalis* (Megastigmidae), *Ormyrus pomaceus* (Ormyridae) and *Torymus auratus* (Torymidae), and details of their assembly can be found in [51]. Supertranscripts were assigned by top score to a taxon of origin, ambiguous sequences were assigned by inspection. We also independently assigned a taxonomic origin to each transcript by aligning to proteins of the 'nr' database (downloaded August 15[th] 2018) with *DIAMOND* [180] and *TAXONOMIZR* (https://github.com/sherrillmix/taxonomizr). The transcriptome was then further filtered to remove fungal contaminant sequences identified as the top-scoring alignment per gene by *DIAMOND*. Ribosomal RNAs were removed after identification by *RNAMMER* and *BLAST* alignment to SILVA large subunit and small subunit rRNA databases [181]. Finally, only transcripts that encode an open reading frame as predicted by *TRANSDECODER* were retained for further analysis. The *SALMON* count matrix was filtered according to the above and split into *Q. robur* gall tissue and normal bud and *B. pallida* specific matrices for differential expression analyses (S5–S7 Files). *BUSCO* (v. 3.0.2) scores were computed for the species-specific transcriptomes against the Eukaryota database (Downloaded July 2017) [182].

## Statistical analysis of stage specific variation in gene expression

Differential expression analysis was performed using *DESeq2* (version 1.6.2) [183] for three separate analyses: gall wasps, oak gall tissues, and normal oak bud and leaf tissues on genes with a greater than 10 combined count across replicates. Count tables and *DESeq2* R scripts

used are provided in S1 File. For gall tissues and gall wasp contrasts a blocking factor for fungi was included where the fungi-derived reads accounted for greater than 1% of the dataset to control for the effect of fungal infection on gall wasp and oak expression (N = 4 samples; S2 Table). A further blocking factor, 'parasitoid load' was fitted for gall wasp transcript analyses, as parasitoid read-depth was greater than 10% of the gall wasp read-depth for nine samples. No blocking factors were fitted for normal bud and leaf comparisons as no fungal or other infection was detected. Samples were inspected for stage-specific clustering by principal components analysis (PCA) after variance-stabilising transformation (VST) of the data in *DESeq2*. Tests were performed sequentially with each stage contrasted with the next developmental stage for galls and normal buds. We specified the 'apeglm' method for effect size shrinkage [184], and imposed a log-fold change requirement of one; meaning that to qualify as differentially expressed, a gene must be expressed significantly greater than log-fold change plus or minus one. This method results in s-values which are analogous to q-values [185], and a cut-off of $\alpha <= 0.005$ was applied as recommended by the 'apeglm' package authors (significant genes: S8–S10 Files). Significant genes were retained for GO term enrichment analysis using the "weight01" algorithm in *TopGO* [186] (Rscript: S1 File) and GO terms annotated by *Trinotate*. Genes of interest were compared against all genes expressed in the analysis, which corresponded to all genes passing initial expression filtering in *DESeq2*. Enriched GO terms were considered significant at a cut-off of $\alpha <= 0.01$. To more easily interpret the results we clustered the resulting GO term lists to reduce redundancy and visualised them in two-dimensional semantic space using *REVIGO* [187]. We performed this for genes that were differentially expressed only in gall tissue contrasts and for genes that were shared and expressed in the same direction between buds (S4 only) and leaves (S5). S9 and S10 Tables for shared genes with concordant expression which corresponds to GO terms enriched for genes corresponding to cells H4:H7 in S4 Table, and for gall tissue-specific genes and GO terms detailed in S5 Table. The *Arabidopsis thaliana* GO term database was used as a reference, 'SimRel' as a similarity measure, the enrichment p-values, and 'medium' similarity. The *B. pallida* Early stage gall-upregulated gene set was compared to all other *B. pallida* genes to identify potential protein motifs associated with induction using MEME [98].

## Orthologue identification of candidate genes

Additional gall wasp genomes were generated for six additional cynipids in six additional tribes within Cynipidae chosen to maximise cynipid diversity analysed according to the phylogeny of Ronquist et al. [54]: *Aylax hypecoi* (tribe Aylacini), *Diastrophus kincaidii* (tribe Diastrophini), *Diplolepis spinosa* (tribe Diplolepidini), *Eschatocerus acaciae* (tribe Eschatocerini), *Pediaspis aceris* (Tribe Pediaspidini) and *Synergus japonicus* (tribe Synergini), and also for two parasitoids in the Figitidae sister group of Cynipidae, *Alloxysta arcuata* and *Parnips nigripes*. Figitidae and Cynipidae are thought to have diverged around 80 million years ago [55], and within the Cynipidae the lineage which diversified into *Diplolepis* and *Pediaspis* split basally from the lineage that includes *Biorhiza* and other oak cynipids [54]. The cynipid and figitid genomes were sampled from entire single individuals and extracted using the DNeasy kit (Qiagen), and Nextera genomic libraries (Illumina) were created using standard protocols. Libraries were sequenced on the Illumina Hi-Seq platform by Edinburgh Genomics. Raw data were quality controlled with *FastQC*, quality and adapter trimmed using *CUTADAPT* and subsequently assembled with *SPAdes* [188] (version 3.5) or *MaSuRCA* [189] assemblers. Assembly quality was assessed by *CEGMA* score. Assemblies were then masked for repeats by a combination of *REPEATMODELER* (v1.0.11) and *REPEATMASKER* (v4.0.6) [190]. Assemblies were checked for contaminant sequences and endosymbionts using a blob-plotting approach [191]

([https://github.com/DRL/blobtools-light](https://github.com/DRL/blobtools-light)). To create blob-plots each contig of the genome was annotated using the DIAMOND aligner in nucleotide *versus* protein search mode with an e-value cut-off of 1e-5, and retaining the five best matches per contig. Coding sequences were then predicted using *AUGUSTUS* (v3.3) with *Nasonia vitripennis* as the reference species [109]. PCWDE enzyme encoding genes were inspected for introns and genes of unambiguously hymenopteran origin in synteny on the same sequence. This would confirm their incorporation into the cynipoid (eukaryotic) genome, rather than being expressed by a previously undetected symbiont. Protein predictions were used as input for *ORTHOFINDER* with default parameters (v2.3.1) to identify orthologue groups [192].

## Supporting information

**S1 File. R scripts used for testing differential expression and GO term enrichment in all comparisons.** Each section is separated by "##" and explanatory text: 1) importing Salmon alignment results and conversion to gene counts, 2) gall wasp larvae *DESeq2* script, 3) gall tissue *DESeq2* script, 4) normal bud tissue *DESeq2* script, and 5) *TopGO* script to identify enriched GO terms. *DESeq2* scripts differ in blocking for fungal infection and parasitoid load in gall wasp tests and fungal infection only for gall tissues.
(TXT)

**S2 File. *Trinotate* pipeline annotation file in excel format.** Annotations for transcripts included in the differential analysis experiments created with the *Trinotate* pipeline. File contains transcripts TRINITY_DN0_c0_g1 to TRINITY_DN63911_c3_g1.
(CSV)

**S3 File. *Trinotate* pipeline annotation file in excel format.** Annotations for transcripts included in the differential analysis experiments created with the *Trinotate* pipeline. File contains transcripts TRINITY_DN63911_c3_g1 to TRINITY_DN76618_c5_g1.
(CSV)

**S4 File. *Trinotate* pipeline annotation file in excel format.** Annotations for transcripts included in the differential analysis experiments created with the *Trinotate* pipeline. File contains transcripts TRINITY_DN76618_c5_g1 to TRINITY_DN236370_c0_g1.
(CSV)

**S5 File. The *Salmon* read-count matrix for the gall wasp (*B. pallida*) data used for differential expression analysis in *DESeq2*.** Each column represents the count for a replicate and each row is a *TRINITY* output component group, taken as a proxy for a candidate gene.
(CSV)

**S6 File. The *Salmon* read count matrix for the oak gall tissue (*Quercus robur*) data used for differential expression analysis in *DESeq2*.** Each column represents the count for a replicate and each row is a *TRINITY* output component group, taken as a proxy for a candidate gene.
(CSV)

**S7 File. The *Salmon* read count matrix for the normal oak bud (*Quercus robur*) data used for differential expression analysis in *DESeq2*.** Each column represents the count for a replicate and each row is a *TRINITY* output component group, taken as a proxy for a candidate gene.
(CSV)

**S8 File. *DESeq2* results for gall wasp genes showing significant differential expression between all gall developmental stages.** The threshold for significance was an S-value

threshold of $< 0.005$ and a $\log_2$-fold change of plus or minus one. Different contrasts are separated by "##" and explanatory text. Columns: 1) gene name, 2) mean expression across all replicates for that gene, 3) log-fold change in base two, 4) log-fold change standard error, and 5) S-value.
(CSV)

**S9 File. *DESeq2* results for oak genes showing significant differential expression between all gall developmental stages.** The threshold for significance was an S-value threshold of $< 0.005$ and a $\log_2$-fold change of plus or minus one. Different contrasts are separated by "##" and explanatory text. Columns: 1) gene name, 2) mean expression across all replicates for that gene, 3) log-fold change in base two, 4) log-fold change standard error, and 5) S-value.
(CSV)

**S10 File. *DESeq2* results for oak genes showing significant differential expression between stages in normal bud development.** The threshold for significance was an S-value threshold of $< 0.005$ and a $\log_2$-fold change of plus or minus one. Different contrasts are separated by "##" and explanatory text. Columns: 1) gene name, 2) mean expression across all replicates for that gene, 3) log-fold change in base two, 4) log-fold change standard error, and 5) S-value.
(CSV)

**S11 File. Gene names for all differentially expressed genes overlapping in S4 Table.** Column headers indicate which cell value the gene list corresponds to in S4 Table reproduced here. Annotations for these genes are provided in S2–S4 Files.
(XLSX)

**S12 File. Gene names for all differentially expressed genes overlapping in S5 Table.** Column headers indicate which cell value the gene list corresponds to in S5 Table reproduced here. Annotations for these genes are provided in S2–S4 Files.
(XLSX)

**S1 Fig. Images of the four replicates of each sexual generation *Biorhiza pallida* gall stage sampled in the RNAseq experiment.** A. Early stage galls. Galls as small as possible were sampled. Bud scales are visible around the galls. B. Growth stage galls. The pink-red epidermis is characteristic of these galls on exposure to sunlight. The lower row comprises cross-sections showing the spongy parenchyma surrounding the larval chambers (up to several hundred in this gall type), which contain small larvae. Vascularisation of tissues can be seen from the point of connection with the oak shoot most clearly in the fourth gall. C. Mature stage galls. The gall epidermis is now brown and papery. The scale bar in all images is in cm/mm.
(EPS)

**S2 Fig. Principle component analysis in *DESeq2* of oak bud developmental stages S1-S4.** X-axis corresponds to principle component one and y-axis to principle component two, and axis labels give the percentage of variance these axes explain in each analysis. A lack of clear separation between stages S0-S4 is observed.
(EPS)

**S3 Fig. REVIGO treemaps of enriched biological process GO terms for oak genes expressed during development of galled and normal oak bud tissues.** These maps were used to construct Fig 5 by providing higher-order groupings of GO terms. Parts A, B, C and D correspond to A, B, C and D in Fig 5.
(EPS)

**S1 Table. *Taxonomizr* classified transcripts of viral origin.** Transcripts were classified as far as the data allow through taxonomic levels from order to species during the annotation process.
(XLSX)

**S2 Table. Read alignment totals for all gall-derived reads.** Data are summarised for each replicate (4 replicates for each of Early, Growth and Mature gall stages), and split into rows for oak (*Quercus robur*), gall wasp (*Biorhiza pallida*), and four major categories of non-target contaminants: fungi, parasitoids excluding *Torymus*, the dominant parasitoid genus *Torymus*, and virus-derived reads.
(XLSX)

**S3 Table. Read alignment totals for oak (*Quercus robur*) normal bud stage replicates.** Data are summarised for each replicate (4 replicates for each of five bud stages S0, S1, S3, S4, and leaf S5) and split into rows for transcripts originating from oak (*Quercus robur*), fungi and viruses.
(XLSX)

**S4 Table. Counts of oak (*Quercus robur*) genes and GO terms that are differentially expressed in both gall and normal bud tissues.** The table gives the total numbers of genes differentially expressed between specific gall developmental stages, and the numbers of these genes that are also differentially expressed between specific stages in normal bud development. The GO term numbers for molecular function (MF) and biological process (BP) categories are the overlap between GO terms enriched for each gall and bud tissue separately. For example, cell H26 represents the overlap in GO terms between genes expressed more highly in Early *versus* Growth stage galls and genes expressed more highly in S4 buds *versus* S5 leaves, in this case an overlap of 27 terms is observed. Data are provided as percentages in an otherwise similar version of the table at right. The direction of differential expression (DE) is given for gall and normal bud tissues. Gall stages compared: E = Early, G = Growth, M = Mature. Normal oak bud stages compared: S0, S1, S3, S4 are bud stages, S5 represents a fully open leaf. Gene names for genes contained in this table are given in S11 File.
(XLSX)

**S5 Table. Counts of oak (*Quercus robur*) genes and GO terms that are only differentially expressed during gall development.** The table gives the numbers of genes differentially expressed between specific gall developmental stages that are <u>not</u> differentially expressed between any stages in normal bud development, and the percentage these genes make up of all differentially expressed oak genes in the given comparison. Direction = in which direction differential expression occurred. Higher is always higher in the earlier developmental stage and *vice versa* for lower. Percentage = percent of total genes or terms that are only differentially expressed in gall tissues for that comparison. Gene names for genes contained in this table are given in S12 File.
(XLSX)

**S6 Table. Normalised counts for specific genes in gall wasp (*Biorhiza pallida*) larvae and oak (*Quercus robur*) gall tissues after differential expression analysis.** The table shows transcript counts across gall stage replicates for specific genes discussed in the text.
(XLSX)

**S7 Table. Oak (*Quercus robur*) biological process (BP) GO terms that were differentially expressed in gall tissues but not during normal bud development.** The table lists the BP GO terms that were up- or downregulated between each pair of gall developmental stages, but

which were not differentially expressed during normal bud development. Annotated = number of genes annotated with that GO term in the complete dataset, Significant = observed number of significantly differentially expressed genes with that annotation, Expected = the expected number of differentially expressed genes with that annotation, P-value = test p-value. We used a significance threshold value of p≤ 0.01.
(XLSX)

**S8 Table. Oak (*Quercus robur*) molecular function (MF) GO terms that were differentially expressed in gall tissues but not during normal bud development.** The table lists the MF GO terms that were up- or downregulated between each pair of gall developmental stages, but which were not differentially expressed during normal bud development. Annotated = number of genes annotated with that GO term in the complete dataset, Significant = observed number of significantly differentially expressed genes with that annotation, Expected = the expected number of differentially expressed genes with that annotation, P-value = test p-value. We used a significance threshold value of p≤ 0.01.
(XLSX)

**S9 Table. Oak (*Quercus robur*) biological process (BP) GO terms that were differentially expressed both in gall tissues and during normal bud development.** The table lists the BP GO terms that were differentially expressed in the same direction between gall developmental stages and between S4 buds and S5 leaves in normal bud development. This is equivalent to GO term enrichment of genes numbered in cells H4:H7 of S4 Table. Annotated = number of genes annotated with that GO term in the complete dataset, Significant = observed number of significantly differentially expressed genes with that annotation, Expected = the expected number of differentially expressed genes with that annotation, P-value = test p-value. We used a significance threshold value of p≤ 0.01.
(XLSX)

**S10 Table. Oak (*Quercus robur*) molecular function (MF) GO terms that were differentially expressed both in gall tissues and during normal bud development.** The table lists the MF GO terms that were differentially expressed in the same direction between gall developmental stages and between S4 buds and S5 leaves in normal bud development. This is equivalent to GO term enrichment of genes numbered in cells H4:H7 of S4 Table. Annotated = number of genes annotated with that GO term in the complete dataset, Significant = observed number of significantly differentially expressed genes with that annotation, Expected = the expected number of differentially expressed genes with that annotation, P-value = test p-value. We used a significance threshold value of p≤ 0.01.
(XLSX)

**S11 Table. Oak (*Quercus robur*) biological process (BP) GO terms that were differentially expressed between stages in normal bud development.** Terms enriched in each comparison and each direction, either up- or down-regulated are included. Annotated = number of genes annotated with that GO term in the complete dataset, Significant = observed number of significantly differentially expressed genes with that annotation, Expected = the expected number of differentially expressed genes with that annotation, P-value = test p-value. We used a significance threshold value of p≤ 0.01.
(XLSX)

**S12 Table. Oak (*Quercus robur*) molecular function (MF) GO terms that were differentially expressed between stages in normal bud development.** Terms enriched in each comparison and each direction, either up- or down-regulated are included. Annotated = number of genes

annotated with that GO term in the complete dataset, Significant = observed number of significantly differentially genes with that annotation, Expected = the expected number of differentially expressed genes with that annotation, P-value = test p-value. We used a significance threshold value of p≤ 0.01.
(XLSX)

**S13 Table. Nucleotide blast results of *Biorhiza pallida* and *Diplolepis spinosa* genomes against *Wolbachia* genomes.** The best match per cynipid contig by bit score is shown and results are ordered by bit score; an e-value threshold of $1x10^{-50}$ was applied. *Biorhiza pallida Wolbachia* contigs have best homology to supergroup A *Wolbachia*, in particular *Wmel* of *Drosophila melanogaster*, whereas *D. spinosa Wolbachia* contigs are most similar to supergroup B *Wolbachia*.
(XLSX)

**S14 Table. Gall wasp (*Biorhiza pallida*) biological process (BP) GO terms that were differentially expressed between stages in gall development.** Terms enriched in each comparison and each direction, either up- or down-regulated are included. Annotated = genes annotated with that GO term in the complete dataset, Significant = observed number of significantly differentially expressed genes with that annotation, Expected = the expected number of genes with that annotation, P-value = test p-value. We used a significance threshold value of p≤ 0.01.
(XLSX)

**S15 Table. *Biorhiza pallida* molecular function (MF) GO terms.** Terms enriched in each comparison and each direction, either up- or down-regulated are included. Annotated = genes annotated with that GO term in the complete dataset, Significant = observed number of significantly differentially expressed genes with that annotation, Expected = the expected number of genes with that annotation. We used a significance threshold value of p≤ 0.01.
(XLSX)

**S16 Table. Intron sequences identified in Cynipid PCWDE genes.** Gene models were predicted using Augustus version 3.3 with *Nasonia vitripennis* as the reference species. The number of genes and number of introns predicted per species is shown for each of the three classes of PCWDE enzyme.
(XLSX)

**S17 Table. OrthoFinder orthogroups of genes more highly-expressed in Early gall larvae with at least one orthologue in another Cynipid species.**
(XLSX)

**S18 Table. Plant Cell Wall Degrading Enzyme Trinotate annotations.** Annotations are given for all transcripts/isoforms that were annotated as a cellulase, pectin/pectate lyase or rhamnogalacturonan lyase.
(XLSX)

**S19 Table. Nucleotide *versus* protein blast results of contigs encoding PCWDEs of bacterial origin or most homologous to PCWDEs present in *D. ponderosae*.** PCWDE sequences for the cynipids *Aylax hypecoi*, *Eschatocerus acaciae* and *Synergus japonicus*, the three most contiguous gall wasp species draft genomes in this study. The table shows best hits for non-overlapping regions of each contig in tabular output format. In addition to annotations for PCWDEs in bacteria and identified as homologous to those in the pine beetle *Dendroctonus ponderosae*, other regions of the same contigs are annotated as genes with homologues in the

Hymenoptera. This confirms that cynipid PCWDE genes are integrated into these species' nuclear genomes alongside ancestrally hymenopteran genes.
(XLSX)

**S20 Table. Combined Illumina read statistics for raw and filtered data for each transcriptome sample.** All sequencing was paired end. Sample Name = original sample identification; Stage = developmental stage of a replicate in gall or bud; Raw Reads = number of read pairs sequenced; Trimmed Reads = number of read-pairs remaining after quality filtering and trimming; Salmon Mapped Reads = number of read pairs mapped by Salmon; Mapping Rate = percentage of trimmed reads that mapped to the transcriptome.
(XLSX)

**S21 Table. Comparison of cynipid venom gland and larval expressed cellulases, showing these to be distinct.** The table shows results of a nucleotide blast comparison between cellulases and chitinases identified in the *Biorhiza pallida* venom gland [49], in larvally expressed transcripts and in the *B. pallida* genome assembly. Results were filtered to included matches with an e-value score of $1e^{-5}$ or less.
(XLSX)

**S22 Table. Mean expression of uniquely DE genes in gall tissue contrasts for gall- and bud tissues.** Genes are sorted by mean expression in the gall comparison, note that mean expression levels are not directly comparable between experiments due to normalisation of gene counts during separate DESEq2 analyses. Where no annotation was made a "." Is present in the description column.
(XLSX)

## Acknowledgments

We thank Jack Schultz for his detailed and insightful feedback on our paper. We also thank Kingston Lacy House, National Trust and Dalkeith Country Park for permission to collect samples at these sites.

## Author Contributions

**Conceptualization:** Jack Hearn, Mark Blaxter, Graham N. Stone.

**Formal analysis:** Jack Hearn.

**Funding acquisition:** Mark Blaxter, Graham N. Stone.

**Investigation:** Jack Hearn, Graham N. Stone.

**Methodology:** Jack Hearn, Mark Blaxter.

**Resources:** Jack Hearn, Mark Blaxter, José-Luis Nieves-Aldrey, Juli Pujade-Villar, Elisabeth Huguet, Jean-Michel Drezen, Joseph D. Shorthouse, Graham N. Stone.

**Visualization:** Jack Hearn, Mark Blaxter, Graham N. Stone.

**Writing – original draft:** Jack Hearn, Mark Blaxter, Graham N. Stone.

**Writing – review & editing:** Jack Hearn, Mark Blaxter, Karsten Schönrogge, Joseph D. Shorthouse, Graham N. Stone.

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
