## [Decision Letter · Decision Letter 0]

30 Sep 2019

Dear Dr Hearn,

Thank you very much for submitting your Research Article entitled 'Genomic dissection of an extended phenotype: oak galling by a cynipid gall wasp' to PLOS Genetics. Your manuscript was fully evaluated at the editorial level and by two of the independent peer reviewers who evaluated the prior related submission. The reviewers appreciated the attention to an important topic but identified some aspects of the manuscript that should be improved.

Both reviewers were complementary of the effort extended to address the prior critiques.  Reviewer #1's comments are primarily focused on the organization of the manuscript and providing the proper context so that readers will be able to understand the importance of the work.  Reviewer #1 offers a lengthy set of suggestions for how to improve the manuscript along those lines.  While many (if not all) of the reviewers comments are potentially helpful, we also understand that you will want the manuscript to reflect your voice and your priorities as authors.  As such, we strongly encourage you to improve the contextualization and logical flow, use standard terminology, and pay very close attention to citing the relevant important literature, but we will leave it up to you to decide how to accomplish that. Reviewer #2's comments can also be addressed by textual/figure revision.  In particular we ask that you pay close attention to Reviewer #2's comments regarding figure clarity, providing sufficient detail in the methods descriptions, and accurately describing the biology of cytokinin production in the system.  The editors anticipate that addressing these concerns will require a significant overhaul of the text and figures, but will not require any additional experimentation.  

We therefore ask you to modify the manuscript according to the review recommendations before we can consider your manuscript for acceptance. Your revisions should address the specific points made by each reviewer.

[LINK]

Yours sincerely,

Gregory P. Copenhaver

Editor-in-Chief

PLOS Genetics

Gregory Barsh

Editor-in-Chief

PLOS Genetics

Reviewer's Responses to Questions

**Comments to the Authors:**

Reviewer #1: This is a newly-submitted manuscript. I reviewed an earlier version, which was also reviewed by two scientists who clearly know much more than I do about genetics, genomics, and molecular techniques. Both reviewers stated that conclusions from the study would be strengthened by the addition of control data showing how oak tissues targeted for gall induction by the insect Biorhiza pallida develop in the absence of the insect. These data have been added in this version, which is a new submission. (I did wonder why the authors changed in the new submission. One former author is now missing?)

The new submission begins with author responses to comments of the three reviewers. My comments were adequately addressed by the authors in this new version.

In hopes that the same two reviewers will be reviewing this new version I leave it to them to comment on the possible limits of this experimental design, which as I understand was to add the requested control plant developmental treatments after having completed the plant-insect treatments earlier. If this is the case, a better experimental design would have been to collect control plant material at the same time as the insect-infested test plant material. I may have misunderstood the design the authors used for this new version.

I will limit my comments to more general topics, specifically revision of the Introduction to make a better argument for their study system. The Abstract similarly needs improvement but I leave it to the authors to do that. At the end of my review, I list specific points that can easily be addressed. In many cases the authors use terminology that is not in common use, being specific to people who study galls.

This is the problem with the Introduction: it fails to make strong arguments for: 1) galls as an interesting biotic interaction in which the host alters its development to accommodate the needs of a symbiont, 2) oak trees as an interesting case of plants that are proficient in making galls, and 3) oak gall wasps as a particularly interesting group of gall inducers. The Introduction should carefully build these arguments. It also needs to make clear what the overall endeavor is and what this particular study achieves for the endeavor.

Many of the comments of the two reviewers seemed to be about the fact that many of the aspects of the interaction were not resolved. The Introduction needs to make clear how complicated the interaction is, not only having many moving parts at any one point in time but also having all these moving parts changing over as the gall and insect develop and mature. This study is an entry point for the endeavor. That should be clear in the Introduction, as should questions that are resolved by this study.

Another problem for the Introduction is the confusion created by presenting a grab-bag of ideas. In many cases, instead of spending some time to explain each idea so that the reader can see its relevance for the experimental work, the idea is simply mentioned and then another idea pops up. One idea the authors seem especially fond of is "extended phenotype". If this is their concept of galls it must be explained. I find its application to plant galls facile or even misleading. The plant makes the gall. It is part of its own phenotype. Moreover, while the concept is perhaps cute for galls made for plant parasites it does not seem to apply for galls that plants make for mutualists like nitrogen-fixing bacteria.

Another idea that is mentioned but not explained is the assertion that understanding galls offer a unique perspective on normal growth and development. Either don't mention this or explain it better. Professor Minelli, an animal developmental biologist, recently turned his attention to plant development biology wrote a book in which he claims that plant galls deserve far more attention as interesting developmental phenomena, independently from the organisms associated with their creation. Minelli pointed to two aspects of plant development that were understudied. Galls were one of them. For me, this idea fits in better with the experimental work in this study.

In terms of the Introduction I found unsatisfying the hypotheses about what galls represent to the plant. Why not just mention these hypotheses in the Discussion? My sense from the manuscript is that the authors know more about insects than plants. I am an insect person rather than a plant person so I cannot tell if the hypotheses they raise about what the gall is from the plant's perspective make sense to a whole organism plant person. Not being a plant person, I wanted an explanation of why plants normally do the two things erected as hypotheses: 1) produce ectopic storage organs and 2) produce somatic embryos. Are these two functions entirely separate or are they simply things that can be identified because of a particular convenient molecular signature? I looked up somatic embryogenesis and it said it is an artificial process in plants not a natural process? And it is not clear how the ENOD system relates to one or both of these plant functions? Or does ENOD only relate to somatic embryos?

Below is an outline of how I would revise the Introduction. Galls are not a subject known by most scientists. Readers who know a little about them see galls as oddities having little relevance for big questions in plant development or host-parasite interactions. An argument must be made.

Right now, the argument seems to be "we have perfect understanding of galls plants make for bacteria but, oh yes, plant make galls for insects too and so let's investigate them too but without referencing what is known about how plants make galls for bacteria." That seems weak to me. As a reader I think: why bother with insect galls? So argue: Insects offer things that bacteria (and fungi) do not. What they offer needs to be made clear. Otherwise readers will think understanding galls made for bacteria and fungi is enough.

Paragraph One: Introduce galls and explain why they are interesting. Make it clear that plants make galls. Say plants make galls in response to many types of foreigners- list them (Redfern 2011 does this). Say plants sometimes make galls for mutualists - explain N-fixing bacteria and fig wasps. Say plants mostly make galls for foreigners that seemingly provide no benefits. We know that many of these no-benefit-for plants foreigners actually harm the plant because, across the vast phylogenetic range of gall inducers, there are many organisms considered to be pests, of agriculture, forestry, and landscapes. The idea is that the foreigner induces a sink at the colonization site and the plant makes a gall there. Resources meant for other plant processes are diverted to the sink. The gall can replace flowers or seeds, thus harming reproduction. Structural problems created at the colonization site can harm plant survival.

Paragraph Two: Use next paragraph to introduce a simple framework and terms that are relevant for your study. Do this by introducing reader to what is known about the galls that are best understood in terms of genetics, including nodules made for mutualist N-fixing bacteria of two types, crown gall made for parasitic Agrobacterium bacteria, and maize smut made for a parasitic fungus Ustilago maydis. Cite major references on these interactions (there are many reviews in Annual Reviews and Nature Reviews). Three important features are 1) release of plant molecules recognized by the foreigner, 2) release of foreigner molecules recognized by the plant, and 3) subsequent changes in plant cellular growth and development that lead to the creation of new accommodations, e.g., a gall, for the foreigner. In the case of the parasites, there are molecules intentionally made in the body of the foreigner and secreted onto or into plant cells that effect a change in host cells that benefits the foreigner. These molecules are called 'effectors'. They are encoded by 'effector genes', candidates of which can be found in the foreigner's genome. Plant molecules that act to recognize foreigner-produced molecules are 'receptors' and are encoded by 'receptor genes', candidates of which can be found in the plant's genome. For both foreigners acting as mutualists or parasites, the plant has growth and developmental traits involved in accommodating the foreigner. They are encoded by various genes found in the plant genome. If the foreigner has the ability to harm the plant, the plant may also have traits that allow it to resist colonization. These are called 'resistance traits' and they are encoded by 'resistance gene', candidates of which can be found in the plant's genome.

Paragraph Three: Explain that, with the exception of root knot and cyst nematodes, understanding of genetics and plant galling responses has lagged behind for insects. And yet plant galls cannot be understood without understanding galls induced for insects. Now make the argument. For example: vast majority of galls made by plants are made for insects, co-evolution and co-speciation, galls made for insects are more elaborate in terms of tissue organization, galls are more species- specific and even sex- or generation- specific, insects interact with plants in different ways to induce galls - stylets, egg-laying females, etc (doubtless the authors are aware of many other reasons, stress ecology and especially evolution). Also, it is interesting that two insect orders with greatest numbers of plant-feeding species have so few species that induce galls.

Paragraph Four: Make the argument for oaks being an interesting plant for galls. Explain the variety of galls oaks make for diverse gall inducers. Is oak a special target of certain types of gall inducers? Explain that genome of oaks was recently published (Plomian et al. 2018- why is this paper not discussed?) What do we learn from oak genome? It has lots of Resistance genes. How does genome make endeavor at hand easier?

Paragraph Five: Make the argument for oak gall wasps (not more generally for cynipids). Explain evolution from parasitoids. Explain relationship between galls and diversification. Explain lineages with and without gall inducers. Make the case for this being an evolutionarily interesting group for plant galls. Explain concept of phylogenetic signal and its relevance for this group of insects (see review in Ann Review Phytopathology). Is this the case for nematodes? I doubt it.

Paragraph Six: Explain exactly how the insect interacts with the plant. Hostfinding. Oviposition and how it sets up idea that female injects something into plant that is first to induce the gall. Do not forget to talk about physical wounding. What happens in plant if female wounds and egg is laid and larva is killed- nothing or the beginnings of the gall? gall made specifically for your gall wasp. Explain you are looking at sexual generation but also explain how this differs for asexual generations. Explain why it is difficult in this system to catch what happens during the very first moments of the larval- plant interaction. Explain if all those larvae developing in the gall are offspring of a single female. Explain how synchronized their development is. Give a figure (a drawing would be best) that shows side by side the development of the larva and the gall (you do not have to show all the larvae). Why does the larva stay so small for so long as the gall first develops?

Paragraph Seven: Explain general methods and objectives for this study. Keep it simple. If you want to get complicated, do it in the Discussion not the Introduction.

I final comment is that often the references are not good choices. In many cases an Annual Review article would be more appropriate than a single research article. Also all the major books on galls should be cited- Redfern, Williams (edited volume) etc.

Simple suggestions:

General: it is often not clear if the term "gall wasp" refers to the species (adults and larvae), the adults, or can also refer to larvae. Specify wasp adult, egg, larva and pupa.

Line 87: define gall induction

Line 108: How many eggs does female lay in this one location and how long does this take? Are all cohabitants of gall siblings? What is known about glands associated with ovipositor in this gall wasp species versus relatives that are not gall wasps?

Lines 108- 131: wounding is not discussed along with many other details relevant to this research (see comments above)

Line 109: "as the egg hatches…" timing of insect versus plant events is not clear- what happens if larvae dies immediately after eclosion?

Line 118: what is size of newly hatched larva relative to larva in Early Stage Galls?

Line 114: explain sexual versus asexual generation

Line 116: not clear - you chose this wasp species because there are lots of larvae inside a single gall?

Line 120: is this usual for early stage larvae to grow very little while the gall itself grows? When do larvae have their growth spurt- when nutritive tissue is first made?

Lines 108-131: is anything known about internal anatomy of first instar larvae, in particular size of salivary glands?

Line 133- 136: It would be nice to have a drawing that shows along one timeline the growth of the larva, the growth of the gall, and the growth of ungalled tissue.

Line 149: the questions could be greatly simplified, 2-3 sentences for each one.

Line 163: "exploited" is a strange word to use for rhizobia which work together with the plant?

Line 170: background for this question should be introduced earlier when explaining oviposition and larval interactions. For each mention how microbial symbionts could be involved?

Line 175: "effector" should be introduced almost at the beginning of the Introduction.

Line 180: not clear who receives the benefit of the microbial symbiont manipulating the plant? All this seems rather vague. Strip it down to basics and discuss in more detail in Discussion.

Line 210: CLAVATA system is mentioned several times in Introduction but we are not told what its relevance is. Why not keep these details for the Discussion?

Line 242: parasitoids were present in the gall? The impact of this on the plant-wasp interaction is not discussed. This should be mentioned in the Introduction. Also gall tissue showed evidence of viral and fungal infections while ungalled tissue did not (Line 676). What does this mean?

Line 296: meaning of term "cecidome" is not explained. Do not use terms like this. Also Line 660.

Line 655: is there a cost to the oak tree of making galls for gall wasps? Discuss the evidence.

Line 716: conclusion that gall is a "novel amalgam" is not really explained. Are the two hypotheses being abandoned?

Line 782: this info should appear in the Introduction

Line 782: in what way are they similar- give an example. Is "many" an exaggeration?

Line 784: this info should be given in the Introduction.

Line 786: "suggesting that the wasp is the source of the galling stimuli" - imprecise use of word "wasp" - be specific about life stage you are talking about.

Line 789: relevance of leaf miners to galls is not clear.

Line 791: section should start with this sentence "We found no support…" the earlier part of this section having been given in the Introduction.

Line 813: effectors should be introduced in Introduction. In Hessian fly, because of gene-for-gene interactions we know the molecules identified as effectors play a role in interactions with plants -read Aggarwal et al. 2014. Genome paper on Hessian fly also provides evidence Zhao et al. 2015.

Line 824: this info about larvae having enlarged salivary glands should be presented in Introduction when effectors are first discussed. Note that Hessian fly larvae also have enlarged salivary glands and much research has been done showing many salivary genes are strongly unregulated during first days of plant interactions. Citing of literature on Hessian fly is spotty.

Line 829: statement "or more generally any genes from other gall-inducers" should have support with references.

Line 841: "On the basis of these data…" this sentence should be the first sentence of the section not the last.

Line 842: I think you mean saliva of gall wasp larvae?

Line 909-915: Discussion of function of PCWDEs seems pretty vague. How about aiding in effectors entering plant cells? What do cytological studies by Bronner indicate is the likely role of these enzymes? Bronner did excellent work and it should not be ignored. Creating a solid story for function of PCWDEs is important if you are going to go on to claim that a PCWDE repertoire is a synapomorphy (please define this term) of gall-inducing cynipids (Line 916)

Line 937-947: you do not really explain the role of ENODs in nodules made for rhizobia and how it might be relevant for gall wasps.

Line 982: Conclusions - Abstract should be more like this - straightforward in terms of what experimental work was done, what were major conclusions, and where the research will go in the future.

Line 996- 1000: this info should be in the Introduction. The reader is left in the dark about so many aspects of this plant-insect system.

Reviewer #2: Summary: This refined manuscript represents a solid omic dissection of gall development by cynipid wasps. The authors have substantially addressed prior reviews to refine the prose, identify specific hypotheses, and correlate (with allowable speculation) changes in plant and insect gene expression at different developmental stages. There remain a few limitations to this study, of which some are explained (not hidden) and thus justifiable, and others require some polishing. There do not appear any lingering major limitations or concerns.

Below are specific comments per section or questions to address:

Response to reviewers.

Overall, the authors did a thorough analysis of the review comments and incorporated changes that should suffice for each reviewer. Having been one of those reviewers (#2), I feel the authors really listened to the criticism and produced an enhanced analysis that reduces speculation and confusion by better correlating their data or revealing new patterns. They explicitly describe the hypotheses being tested, the limits of their data, and how these results advance the literature.

abstract.58. if the comparison includes galling, inquiline, and non galling (non herbivore) species, could the authors extrapolate that PCWDEs are one path to herbivory, given several other herbivores evolved these. Hessian fly, another gall did not have these. The fig wasp does not appear to either. Thus maybe these enzymes are not key to galling, unless you delineate it as key to cynipid wasp galling

Wybouw et al 2016 GBE discusses PCWDE from herbivores if it helps

author.summary.66. this seems a strong statement, assuming the interaction must either negatively affect the host or benefit both. There are many cases where gall formation does not induce fitness costs, so while the insect may live on the plant, reorganizing sink and other signaling may enhance resource acquisition. Perhaps the authors would change this to "The organisms involved range from mutualistic to parasitic" to better encompass the gradient of host interactions.

intro.81. "interactions ranging from mutualist to parasitic"

99. "gall communities contribute significantly to biodiversity" of what? of species in general? of gall species? this is unclear.

108. what is a generation specific tissue? does this mean the generation of the insect or plant?

136. thus, cynipids are maintaining a meristem or delaying its maturity. It is interesting to think about the mechanism for this.

178. “some gall-inducing insects have been shown to produce endogenous auxins and/or cytokinins (see below).”

208. “Endogenous production of plant hormones (such as auxin and cytokinins) has been shown in a range of galling and non-galling herbivorous insects [],” In the first sentence the authors indicate insects produce these compounds (but do not differentiate if their co-bionts are responsible). In the later sentence the presence of these compounds is indicated (not the production) suggesting a co-biont may be responsible (even if this isn’t stated). The authors would be best served by removing the misleading intimation in the first sentence that insects synthesize cyotkinins until data show up that substantiate this claim.

This point was raised in the review and the authors claimed to have corrected it in the response to reviewers. Indeed, the discussion reads as if the hypothesis that insects synthesize cytokinin remains possible yet unvalidated, although evidence thus far provides no support, questioning if it should still remain a viable hypothesis. In the intro, I would expect a similar statement, and not one that purports that some galling insects produce cytokinins. This paper would benefit (and be more cited) if they recognized this throughout all sections.

226. what is the evolutionary age of the split? for insect feeding parasitoids (highly specialized) and plant feeders (highly specialized) the divergence may be so long that rapidly evolving genes underlying a potentially antagonistic interaction (assuming the most negative of parasitism) diverged quickly so signatures or orthology are so low they may be missed. Thus, some caution should be used in this comparison.

results.

table.3. can the numbers be parsed into up(down) or something that incorporates more context?

b. what is a "number of DE genes that were not DE between any stages in normal bud development" could this be "unique DE genes in galls not found in DE of normal bud development"

c. this is confusing. is this table the same as saying 1(26) up(down) in S0 vs S1? if so, using this notation would decrease the table size. It seems intuitive if in a pairwise comparison you have up and down genes, then the higher expression will be based on the direction of the comparison where up are genes up in 0 vs 1 and down are genes up in 1 vs 0.

416 may be constitutively expressed. did you check counts?

462 if no JA response, how antagonistic is this interaction? This is in reference to what level of negative response defines a parasite vs a commensal.

488 how was the microbial profiling done?

524. yes constitutive expression may prevent detection.

570-572. this is not surprising given the lineage specificity of genes involved in plant manipulation (eg., Boulain et al 2018)

table.4 can you put a dated phylogeny on the edge of the table so we know how different these insects are?

606. what about the inquiline? in table 5 the non galler retains PCWDEs indicating it may also be important for herbivory but not galling

table.5 can this be in the same order as table 4.

830. yes see boulain reference above

874. yes this is how to discuss cytokinins. why is the intro set up differently?

924. maybe the step was not direct to galling but to herbivory.

methods.

where is the analysis for detecting microbial sequences? STable 13 suggests you only blasted your assembly against Wolbachia, but did not survey for other microbes.

fig.2. This could be clearer, 1) perhaps with reduced text and better delineation of which organism’s genes you are making the prediction on. 2) seems like export through active secretion could apply to any organism (insect or microbe) living in the gall and not just the insect. 3) hypotheses are not true, just supported, so perhaps reword this in the caption. 4) Perhaps indicate A represents the expected gene expression patterns if the induce phenotype results in an ectopic storage organ or a modified somatic embryo. 5) counting up now I expect 3 to be similar in concept to 2, but it is not because 1&2 represent the expectations of the plant transcriptome and 3-5 represent what genes are expressed that are not of plant origin. I don't think the number should be continuous between letters. 6) Perhaps change the title from a question to something that captures all the boxes are transcriptome patterns, ie “Expected (or hypothetical) gene expression patterns for galls and associated gallers during development” 7) export is repeated a lot so maybe use an arrow? 8) why are processes different from stimuli – it seems a secretory protein, enzyme, or rna could easily result in the alteration of gene expression of processes. How these are different is unclear.

fig.4. This is not clear. I gather that 747 genes are up in early vs growth and 2310 are up in S4 vs S5 of buds, but what is not clear is the 311 that increased in both gall and normal buds. Are you saying that 311 of the same genes were up in the first and latter comparison I just described (as a venn diagram would indicate). This would suggest 311 belong in the set of 747 and of 2310. If you wanted to make more contrast between the genes that are unique you could label circles as “unique genes” found DE for the specific comparison and boxes as “shared” genes. This would result in a circle of 436 unique to early vs growth stage galls and 1999 unique to normal bud S4 vs S5. I think identifying what is unique is a stronger result than combining and having the reader subtract to get the number. Total gene numbers is more insightful than percents, and in an already complex figure the fewer the numbers the better. You could also present this in a more standard venn diagram way with overlapping circles. If my interpretation of this graph was wrong, then you may need to reconsider it completely, because it is very complex.

fig.5. is this to be oriented differently in the manuscript? Fig4 is aligned well.

**Have all data underlying the figures and results presented in the manuscript been provided?**

Reviewer #1: Yes

Reviewer #2: Yes

PLOS authors have the option to publish the peer review history of their article (what does this mean?). If published, this will include your full peer review and any attached files.

Reviewer #1: No

Reviewer #2: No

---

## [Editor Report · Decision Letter 1]

23 Oct 2019

Dear Dr Hearn,

We are pleased to inform you that your manuscript entitled "Genomic dissection of an extended phenotype: oak galling by a cynipid gall wasp" has been editorially accepted for publication in PLOS Genetics. Congratulations!

Yours sincerely,

Gregory P. Copenhaver

Editor-in-Chief

PLOS Genetics

Gregory Barsh

Editor-in-Chief

PLOS Genetics

Comments from the reviewers (if applicable):

**Data Deposition**

http://datadryad.org/submit?journalID=pgenetics&manu=PGENETICS-D-19-01463R1

Press Queries

---

## [Editor Report · Acceptance letter]

28 Oct 2019

PGENETICS-D-19-01463R1 

Genomic dissection of an extended phenotype: oak galling by a cynipid gall wasp 

Dear Dr Hearn, 

We are pleased to inform you that your manuscript entitled "Genomic dissection of an extended phenotype: oak galling by a cynipid gall wasp" has been formally accepted for publication in PLOS Genetics! Your manuscript is now with our production department and you will be notified of the publication date in due course.

With kind regards,

Matt Lyles

PLOS Genetics

On behalf of:
